# The first complete 3D reconstruction and morphofunctional mapping of an insect eye

Anastasia A Makarova[1], Nicholas J Chua[2], Anna V Diakova[1], Inna A Desyatirkina[1], Pat Gunn[2], Song Pang[3,4], C Shan Xu[3,5], Harald F Hess[3], Dmitri B Chklovskii[2,6], Alexey A Polilov[1]*

[1]Department of Entomology, Faculty of Biology, Lomonosov Moscow State University, Moscow, Russian Federation; [2]Center for Computational Neuroscience, Flatiron Institute, New York, United States; [3]Janelia Research Campus, Howard Hughes Medical Institute, Ashburn, United States; [4]Yale School of Medicine, New Haven, United States; [5]Department of Cellular and Molecular Physiology, Yale School of Medicine, New Haven, United States; [6]Neuroscience Institute, NYU Langone Medical Center, New York, United States

## eLife Assessment

This **valuable** study sets new standards in analyzing the ultrastructure of insect eyes, which have long served as models for understanding how vision works. The way it describes an entire eye with the resolution of electron microscopy is **convincing**. On top of this, a miniaturized visual system provides additional, remarkable insights towards understanding optimized solutions.

*For correspondence:
polilov@gmail.com

**Abstract** The structure of compound eyes in arthropods has been the subject of many studies, revealing important biological principles. Until recently, these studies were constrained by the two-dimensional nature of available ultrastructural data. By taking advantage of the novel three-dimensional ultrastructural dataset obtained using volume electron microscopy, we present the first cellular-level reconstruction of the whole compound eye of an insect, the miniaturized parasitoid wasp *Megaphragma viggianii*. The compound eye of the female *M. viggianii* consists of 29 ommatidia and contains 478 cells. Despite the almost anucleate brain, all cells of the compound eye contain nuclei. As in larger insects, the dorsal rim area of the eye in *M. viggianii* contains ommatidia that are believed to be specialized in polarized light detection as reflected in their corneal and retinal morphology. We report the presence of three 'ectopic' photoreceptors. Our results offer new insights into the miniaturization of compound eyes and scaling of sensory organs in general.

## Introduction

Sensory organs are essential for informing the animal about its environment and thus for its survival. The study of these organs has revealed important biological principles in insects (*Borst and Egelhaaf, 1989*; *Strausfeld, 1989*; *Warrant and McIntyre, 1993*; *Meyer-Rochow, 2015*; *Meyer-Rochow and Gál, 2004*; *Warrant and Nilsson, 2006*; *Borst, 2009*). The study of the principles of the scaling in sensory organs is a highly interesting and complex objective for morphology and bionics (*Srinivasan et al., 1999*; *Graham and Philippides, 2012*; *Serres and Viollet, 2018*). Most ultrastructural studies on insect sensory organs are currently performed using traditional scanning EM (external morphology)

and/or single-section transmission EM (internal ultrastructure). Because these methods involve scanning only parts of the sensory organs, they do not provide a comprehensive perspective. Modern methods of volume electron microscopy (vEM) allow studying the structure of insects at the ultrastructural level (*Meinertzhagen, 2018*; *Kawasaki et al., 2019*). However, the peculiar constructive features of these methods often limit the minimum size of the studied organisms (*Xu et al., 2017*). The methods of vEM require complex material staining, much time for scanning, and a long period for proofreading (*Plaza et al., 2014*), and as a result reconstructions of whole organisms (whole-body connectomics) using vEM are rather scarce and deal only with a few animals (*White et al., 1986*; *Varshney et al., 2011*; *Ryan et al., 2016*; *Cook et al., 2019*; *Jékely et al., 2024*).

Minute insects are convenient and interesting organisms for study. They have many physiological, cognitive, and behavioral capacities found in larger insects, while their small body sizes make it possible to reconstruct in detail not only their sensory organs (*Diakova et al., 2022*), but also whole organ systems (*Desyatirkina et al., 2023*). Their small sizes also permits tracing of the complete pathways that connect their sense organs with particular regions of the brain (*Chua et al., 2023*). Three-dimensional (3D) reconstructions of sensory organs make it possible to assess and test the data obtained earlier using traditional electron microscopy, as well as specification of the spatial orientation, shape, and volumetric parameters of cells and organelles. Three ommatidia of a compound eye were reconstructed in 3D for the first time in the parasitoid wasp *Trichogramma evanescens* (*Fischer et al., 2019*), a study that broadened the notions of the functional and structural limits in miniaturized sense organs. In spite of the progress in methods of electron microscopy, until now a detailed 3D description of a complete compound eye has never been published.

The peculiar features of structure and function of compound eyes, ultrastructure of photoreceptors, and the associated functional and structural limits of insects associated with small body sizes were studied in detail in several insect orders: Coleoptera (*Meyer-Rochow and Gál, 2004*; *Makarova and Polilov, 2018*; *Makarova et al., 2019*), Hymenoptera (*Fischer et al., 2011*; *Makarova et al., 2015*; *Palavalli-Nettimi and Narendra, 2018*; *Fischer et al., 2019*; *Palavalli-Nettimi et al., 2019*), Diptera (*Meyer-Rochow and Yamahama, 2019*), and Lepidoptera (*Honkanen and Meyer-rochow, 2009*; *Fischer et al., 2012*; *Fischer et al., 2010*, , *Fischer et al., 2014*); for review, see *Makarova et al., 2022a*, *Makarova et al., 2022b*. Despite the considerable progress in the study of the miniaturization

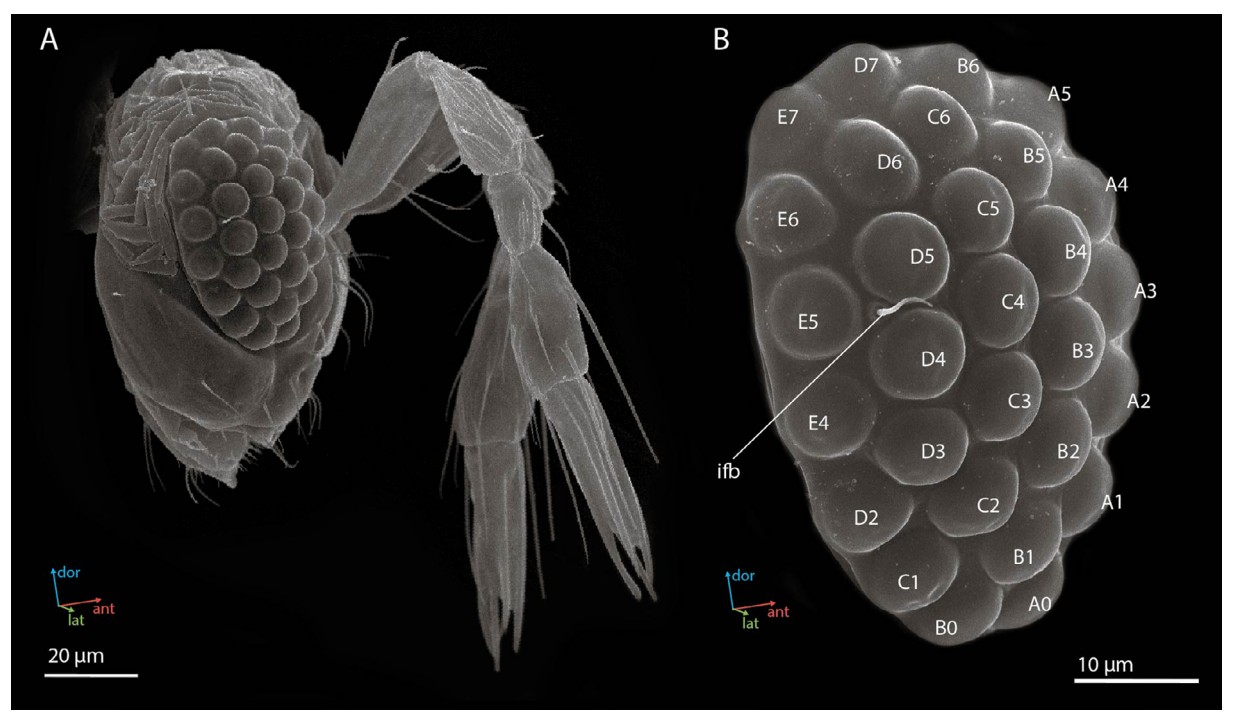

**Figure 1.** Scanning electron microscopy (SEM) images of the head (**A**) and the compound eye (**B**) of a female *Megaphragma viggianii* (side view). Ifb, interfacet bristle. The compound eye comprises 29 ommatidia named here as in *Chua et al., 2023*.

of compound eyes, some issues remain incompletely investigated (e.g., the structure and spatial position of cells and subcellular elements of the eye or the functional specialization of ommatidia).

*Megaphragma viggianii* is a minute parasitic wasp. It is one of the smallest known species of the family Trichogrammatidae and an egg parasite of thrips (*Bernardo and Viggiani, 2002*). Three species of *Megaphragma* are known to display the phenomenon of the lysis of nuclei in neurons during later periods of pupal development (*Polilov, 2012*; *Polilov, 2017*; *Makarova et al., 2022c*). The general structural organization of compound eyes and adaptations associated with miniaturization were described on the basis of single EM sections in *M. polilovi* (*Makarova et al., 2015*). Data on the optical properties of the ommatidia and the connectome of the first visual neuropil (the lamina) were first obtained using the full dataset of a *Megaphragma* head (*Chua et al., 2023*).

The main goal of this study is to reveal the ultrastructural organization of compound eyes in the microinsect *M. viggianii*. Complete cellular reconstruction of compound eyes using vEM based on focused ion beam (FIB) SEM makes it possible to visualize the 3D structure of the whole eye and to trace the pathways that connect the detector with the brain. Morphometric analysis, in turn, makes it possible to quantify the number and cellular composition of ommatidia and assess the sizes and volumes of subcellular elements, providing data on the scaling of sense organs.

## Results

### General description of the compound eye

Female compound eyes of the parasitoid wasp *M. viggianii* are oval in shape (*Figure 1A*) and measure about 50.6±1.5 µm (hereinafter mean ± s.d.) in dorsoventral extent. Their anterior-posterior extent is on average 32.6±0.73 µm. Each eye has 29 facets. The corneal surface of the facets is smooth. A single interfacet bristle is present near the posterior row of the eye, between facets E5, D5, and D4 (*Figure 1B*). No differences are visible on SEM (at the external cuticular level) between the facets of the eye.

### General description of ommatidia

Using the vEM of the whole *M. viggianii* eye we found a total of 478 cells: 261 photoreceptor cells, 116 cone cells, 58 primary pigment cells (PPCs), 24 secondary pigment cells (SPCs), 16 rim pigment cells (surrounding the eye on the periphery), and 3 'ectopic' photoreceptors (see 'Ectopic' photoreceptors). Each of the 29 ommatidia contains nine photoreceptor cells, four cone cells, and two PPCs (*Figure 2A*; see *Video 1*). The length of one ommatidium is on average 21.2 µm, gradually increasing from the dorsal rim area (DRA) to the ventral part of the eye (*Table 1*).

### The dioptric apparatus

The dioptric apparatus (DA) of each ommatidium consists of the biconvex lens and crystalline cone (*Figures 2D, E and 3A, G*). The diameter of the lens is 6.9±0.89 µm in DRA and 8.0±0.77 µm in non-DRA ommatidia (*Table 1*). The lenses are covered by a cuticle of 0.28±0.050 µm depth (*Figure 3A and G*). The greatest thickness of the lens is 2.5±0.56 µm in DRA and 3.3±0.40 µm in non-DRA ommatidia. The outer/inner radii of lens curvature are 3.3±0.64/1.2±0.32 µm and 4.7±0.39/3.0±0.52 µm in DRA and non-DRA ommatidia, respectively. The volume of the lens is 16.5±14.0 µm³ in DRA and 41.4±9.0 µm³ in non-DRA (*Table 2*; see *Supplementary file 1b*).

The crystalline cone comprises four cone cells (*Figures 2D, E, 3B, C, H, I, and 4A, C*). Each cone cell has a long and thin projection that extends down to the basal matrix along retinal cells (*Figure 3D–F and J–I*). Due to the constant position of cone cell projections, we enumerate them according to their passing between the retinula cells: C1: between R1 and R1; C2: between R3 and R7'; C3: between R4 and R5; C4: between R6 and R7(R8) (see 'Retinula cells and rhabdom') (*Figure 3D–F, J, and K*). DRA ommatidia have small cones with nuclei that fill most of the cone cell volume (*Figures 2D, E, 3A–C, and 4A, B, E, F, I, J*). In DRA ommatidia, the mean volume of a cone cell is 13.1±6.0 µm³ and of its nucleus 6.6±0.59 µm³ (*Table 2*). In non-DRA ommatidia, the nuclei are elongated, positioned in the upper third of the cells perpendicular to the ommatidial long axis, leaving the central part of the cone free (*Figures 2D, E, 3G–I, 4C, D, G, H, K, L, and 5G–I*). Reconstruction has shown that the nuclei of non-DRA ommatidia form an aperture (*Figures 2D and 4D, H*). The mean volume of a cone cell in non-DRA ommatidia is 32.9±5.9 µm³ and of its nucleus 7.5±0.93 µm³ (*Table 2*).

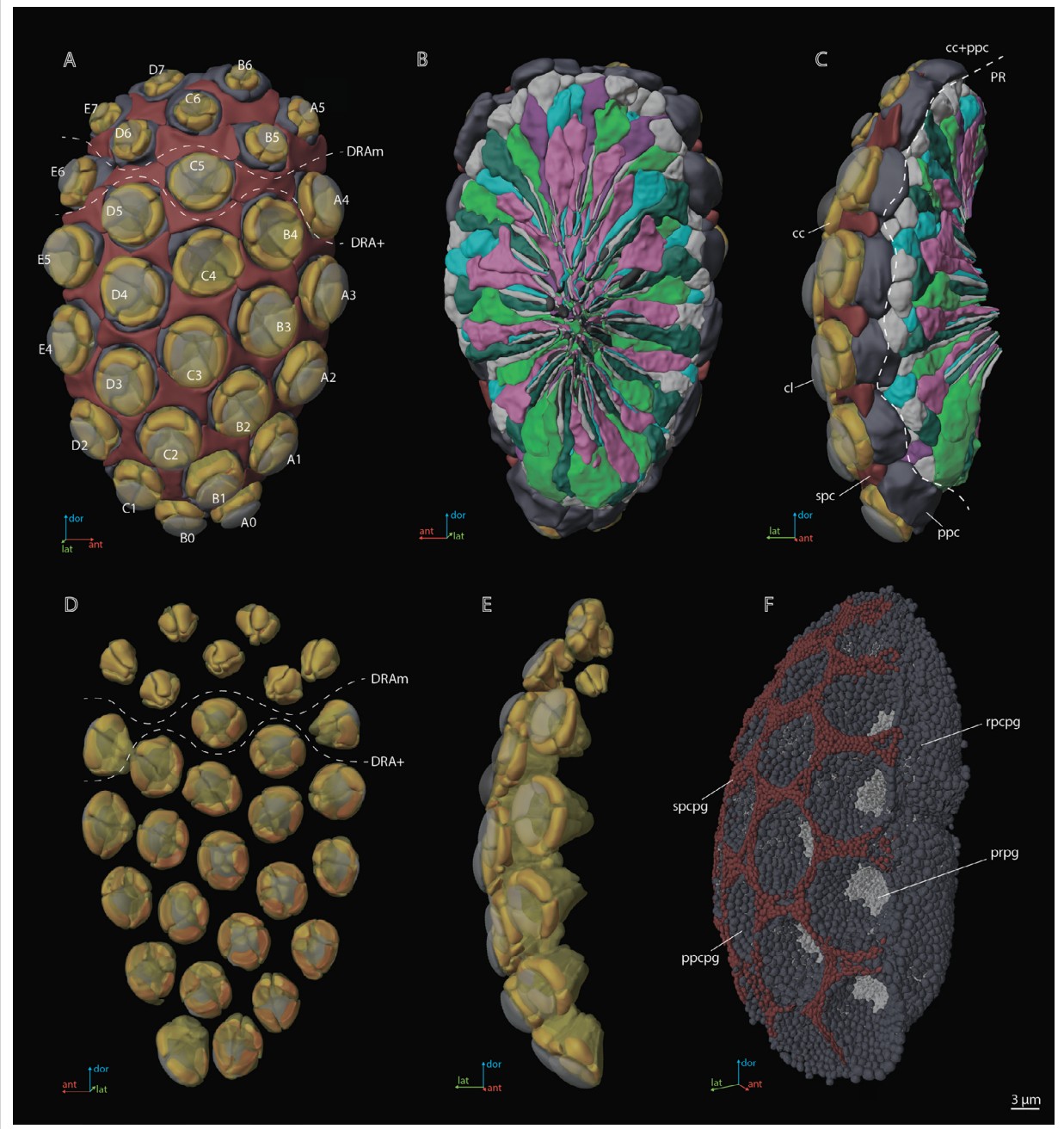

**Figure 2.** A three-dimensional (3D) reconstruction of the compound eye of *M. viggianii*. (**A**) A front view from the cornea side; (**B**) a rear view from the retinal side; (**C**) a side view; (**D**) a rear view of the ommatidia DA; (**E**) a rear view of the ommatidia DA; (**F**) a semi-side view of pigment granules of all cells. cc, crystalline cones; cl, corneal lense; DRAm, dorsal rim area ommatidia (morphological specialization); DRA+, transitional zone ommatidia; ppc, primary pigment cells; ppcpg, pigment granules of PPC; prpg, retinal (photoreceptor) pigment granules; rpcpg, rim cells pigment granules; spc, secondary pigment cells; spcpg, secondary pigment cell pigment granules. Ommatidia are named as in *Chua et al., 2023*.

## Primary pigment cells

Two PPCs envelop the cone of each ommatidium (*Figures 2C, 3, and 4*) and are situated lower than the SPCs. The volume of PPC is 47.1±14.4 µm$^3$ in DRA ommatidia and 64.5±11.4 µm$^3$ in non-DRA ommatidia (*Table 2*). The PPCs are densely filled with spherical pigment granules, identical in DRA and non-DRA ommatidia (*Figures 2F, 3, and 5*). The granules have a mean volume of 0.18±0.039 µm$^3$. The PPCs contain on average 144±52 pigment granules, their total volume per cell being 20.63±10.52 µm$^3$. The nuclei are positioned in the lower half of the cells, beneath the level of the SPCs

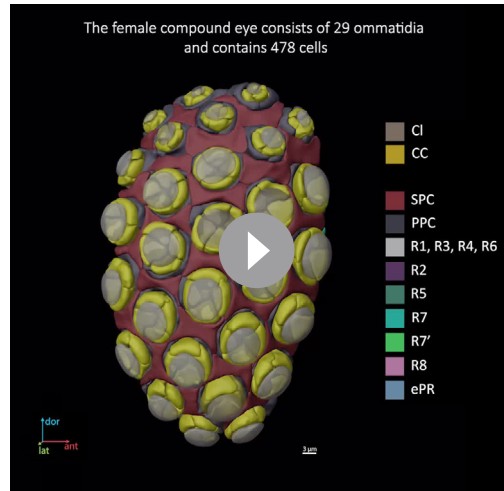

The female compound eye consists of 29 ommatidia and contains 478 cells

Legend:
- CI
- CC
- SPC
- PPC
- R1, R3, R4, R6
- R2
- R5
- R7
- R7'
- R8
- ePR

**Video 1.** 3D reconstruction of the ultrastructure of the compound eye of *M. viggianii*. cc, crystalline cones; cl, corneal lense; ccn, nuclei of crystalline cone cells; ePR, 'ectopic' photoreceptors; ppc, primary pigment cells; rbd, rhabdoms; spc, secondary pigment cells; R1–R8, retinal cells.

https://elifesciences.org/articles/103247/figures#video1

(*Figure 4*). The volume of PPC nuclei is equal in DRA and non-DRA ommatidia; their mean volume is 7.5±0.94 μm³ (7.5±0.86 μm³ and 7.6±0.98 μm³ for DRA and non-DRA, respectively) (*Table 2*). Several small oval mitochondria are positioned in the dorsal half of PPC (*Figure 6*). The mean chondriome volume is 0.46±0.14 μm³.

## Secondary pigment cells

Twenty-four SPCs are positioned directly beneath the cornea (*Figures 2–4*). Each ommatidium of the central part of the eye is surrounded by four SPCs, while each marginal ommatidium is surrounded by two SPCs, on the internal margin of the eye (*Figure 2A*). No extensions of SPC adjoin the retinula cells down to the basal matrix. The volume of SPC is similar near DRA and non-DRA ommatidia (*Table 2*). The nuclei of SPC have a mean volume of 6.4±1.0 μm³. The SPCs are filled with pigment granules (*Figures 2F and 5B, D*), 158±28 per cell, having a mean unit volume 0.050±0.016 μm³. The total volume of pigment granules per cell is about 7.9±3.0 μm³ (*Table 3*). The shape of the granules of SPC in the dorsal third of the eye, near the DRA ommatidia, is round. The shape of pigment granules in the center and proximal third of the eye (around the non-DRA ommatidia) is oval (*Figure 5B and D*). Several small oval mitochondria are positioned in the dorsal half of SPC (*Figure 6*). The mean volume of the chondriome is 0.24±0.15 μm³.

## Retinula cells and the rhabdom

The retina area of each ommatidium consists of nine photoreceptor cells (PR) (*Figures 3 and 4*; *Figure 3—figure supplements 1–5*), six of which (R1–R6) send short axons that project to the lamina and the remaining three (R7, R7', R8) send long axons that reach the medulla. The position of the eighth retinula cell in relation to the position of the cone cell projections and the axon targets in the optic lobes can be used for recognition and labeling of all other cells (see 'Identifying the retinula cells and terminology'). The nuclei of retinula cells are arranged on four levels (*Figure 4*). The most distal position is occupied by the nuclei of PR R1, R3, R4, and R6. More proximally the nuclei of partner cells R2 and R5 are situated opposite each other and R7 PR. The nuclei of R7' and R8 PR are positioned proximally; R8 is the lowest (*Figure 4*). The majority of R7 cells show lighter rhabdomeres (less electron density) than other PR (*Figure 3D and J*; *Figure 3—figure supplements 1–5*).

Nine retinula cells form the fused rhabdom of *M. viggianii* (*Figure 3*). The rhabdomere of R7 is replaced by the rhabdomere of R8 approximately in the center of ommatidium length in non-DRA

**Table 1.** Linear measurements (μm) of *M. viggianii* eye components.

Diameter[*,†], diameter of rhabdom measured in orthogonal planes, according to its not round shape. Hereinafter mean ± s.d. DRA, dorsal rim ommatidia in general (DRAm and DRA+); DRAm, dorsal rim area ommatidia (morphological specialization); DRA+, transitional zone ommatidia; non-DRA, regular (non-DRA) ommatidia. Raw data, *Supplementary file 1a*.

| | | Lense | | | | Cone | | Rhabdom | | | | |
|---|---|---|---|---|---|---|---|---|---|---|---|---|
| | | | | Curvature | | | | | | | | |
| | Ommatidium length | Diameter | Thickness | inner | outer | Length | Width | Diameter (distal) | Diameter central[*] | Diameter central[†] | Diameter mean | Length |
| DRA | 19.2±0.37 | 6.9±0.89 | 2.5±0.56 | 1.2±0.32 | 3.3±0.64 | 3.0±0.38 | 4.9±0.66 | 2.0±0.16 | 2.1±0.32 | 2.2±0.23 | 2.1±0.18 | 13.4±0.64 |
| DRAm | 19.0±0.20 | 6.5±0.51 | 2.2±0.28 | 1.1±0.21 | 3.1±0.51 | 2.8±0.28 | 4.5±0.21 | 1.9±0.11 | 1.9±0.16 | 2.2±0.24 | 2.1±0.11 | 13.7±0.29 |
| DRA+ | 19.7±0.15 | 7.9±0.85 | 3.2±0.26 | 1.6±0.19 | 3.9±0.68 | 3.4±0.14 | 5.8±0.21 | 2.2±0.068 | 2.5±0.21 | 2.2±0.25 | 2.4±0.11 | 12.6±0.63 |
| Non-DRA | 22.3±2.5 | 8.0±0.77 | 3.3±0.40 | 3.0±0.52 | 4.7±0.39 | 4.6±0.60 | 6.7±0.25 | 2.7±0.26 | 2.9±0.25 | 2.8±0.27 | 2.9±0.21 | 14.2±1.8 |

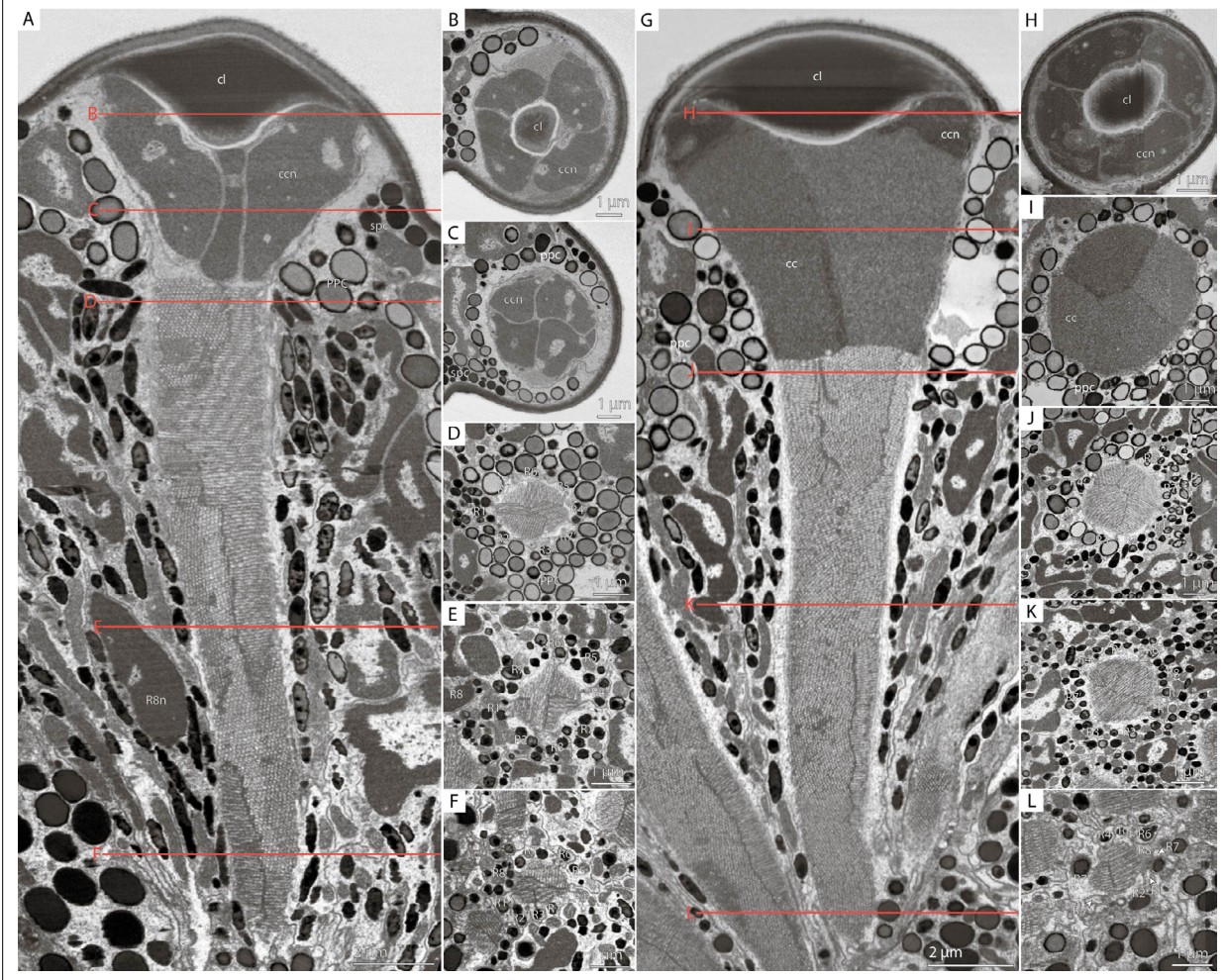

**Figure 3.** Cross-sections of *M. viggianii* ommatidia sampled from a volume electron microscopy (vEM) (FIB-SEM) dataset. (**A–F**) Dorsal rim area (DRA) ommatidia (B5); (**G–L**) non-DRA ommatidia (C3). (**A, G**) A longitudinal section through one ommatidium; (**B, H**) a cross-section through the proximal part of a corneal lens; (**C, I**) a cross-section through the center of a cone; (**D, J**) a cross-section through a distal rhabdom, directly under the cone; (**E, K**) a cross-section through the center of a rhabdom; (**F, L**) a cross-section through a distal rhabdom. cc, crystalline cone; ccn, nuclei of crystalline cone cells; cl, corneal lens; ppc, primary pigment cells; R1–R8, retinal cells; spc, secondary pigment cells; asterisk (*) marks cone cell projections.

The online version of this article includes the following figure supplement(s) for figure 3:

**Figure supplement 1.** Scheme of the compound eye of *Megaphragma viggianii* (A); Scheme of cross-sections of ommatidia (B); EM sections through an ommatidium of *M. viggianii* (C).

**Figure supplement 2.** EM sections through an ommatidium of *M. viggianii*, continuation of *Figure 3—figure supplement 1* (C).

**Figure supplement 3.** EM sections through an ommatidium of *M. viggianii,* continuation of *Figure 3—figure supplement 1* (C).

**Figure supplement 4.** EM sections through an ommatidium of *M. viggianii*, continuation of *Figure 3—figure supplement 1* (C).

**Figure supplement 5.** EM sections through an ommatidium of *M. viggianii*, continuation of *Figure 3—figure supplement 1* (C).

ommatidia (*Figure 3K*) and in the proximal third in DRA ommatidia (*Figure 3F*; see *Figure 3—figure supplements 1–5*). The distal diameter (under the cone) is 2.0±0.16 µm in DRA ommatidia and 2.7±0.26 µm in non-DRA ommatidia. The cross-section of the distal rhabdom in DRA ommatidia is nearly oval in the first half of the rhabdom, and becomes rectangular in the center of the ommatidium along its length (*Figure 3D–F*; see *Figure 3—figure supplements 1–5*, *Figure 2* of DRA ommatidia). The cross-section of non-DRA ommatidia has a circular shape (*Figure 3J–L*; see *Figure 3—figure supplements 1–5*, *Figure 2* of non-DRA ommatidia). The orientations of microvilli in long photoreceptor cells (R7 and R7') of DRA ommatidia are orthogonal to each other and consistent throughout

**Table 2.** Mean volumes ($\mu m^3$) for cellular and subcellular elements of ommatidia in *M. viggianii*. The volumes were obtained from 3D models. DRA, dorsal rim area ommatidia (morphological specialization); DRA+, transitional zone ommatidia; non-DRA, regular (non-DRA) ommatidia; R1-R8, retinal cells; PPC, primary pigment cells; SPC, secondary pigment cells. Raw data, *Supplementary file 1b*.

| | | Retinal cell | | | | | | | | | | | | |
| | | R1 | R2 | R3 | R7' | R4 | R5 | R6 | R7 | R8 | Cone | PPC | SPC | Lense |
|---|---|---|---|---|---|---|---|---|---|---|---|---|---|---|
| Soma | DRA | 17.5±2.0 | 24.4±3.8 | 17.9±2.3 | 33.0±2.5 | 18.0±2.0 | 25.2±4.1 | 17.9±1.6 | 32.1±4.8 | 15.8±1.9 | 13.1±6.0 | 47.1±14.4 | 22.0±4.7 | 16.5±14.0 |
| | DRAm | 16.4±1.2 | 22.3±2.3 | 17.7±2.3 | 32.2±2.0 | 17.2±1.4 | 23.4±3.1 | 17.1±0.8 | 30.0±3.4 | 14.8±0.8 | 9.5±1.2 | 40.6±9.5 | 20.2±4.3 | 8.6±3.3 |
| | DRA+ | 19.8±2.7 | 29.2±3.7 | 18.3±3.1 | 34.8±3.7 | 20.0±3.0 | 29.4±4.3 | 19.9±1.9 | 37.2±6.7 | 18.2±2.5 | 21.3±4.5 | 61.4±13.0 | 26.2±2.6 | 34.8±17.2 |
| | Non-DRA | 21.0±3.3 | 29.5±3.1 | 22.0±2.6 | 53.3±11.1 | 22.4±3.5 | 31.1±3.2 | 20.5±4.5 | 30.6±8.5 | 28.2±4.3 | 32.9±5.9 | 64.5±11.4 | 24.5±3.2 | 41.4±9.0 |
| Nuclei | DRA | 6.5±0.46 | 7.2±0.89 | 6.7±0.64 | 8.2±0.83 | 6.7±0.52 | 7.4±0.73 | 6.8±0.36 | 8.4±1.03 | 7.0±0.63 | 6.6±0.59 | 7.5±0.86 | 6.6±0.78 | n/a |
| | DRAm | 6.4±0.42 | 6.8±0.72 | 6.8±0.62 | 8.4±0.61 | 6.7±0.53 | 7.3±0.66 | 6.8±0.38 | 8.3±0.98 | 6.8±0.53 | 6.5±0.65 | 7.4±0.93 | 6.8±0.76 | n/a |
| | DRA+ | 6.8±0.73 | 8.0±1.2 | 6.4±0.75 | 7.6±1.1 | 6.8±0.60 | 7.7±0.72 | 6.9±0.47 | 8.6±0.87 | 7.4±0.73 | 6.9±0.38 | 7.5±0.75 | 5.9±0.42 | n/a |
| | Non-DRA | 7.0±0.56 | 7.6±0.91 | 7.2±0.73 | 9.2±1.1 | 7.2±0.85 | 7.9±0.71 | 6.9±0.98 | 7.9±1.2 | 8.3±0.80 | 7.5±0.93 | 7.6±0.98 | 6.4±1.0 | n/a |
| Rhabdomere | DRA | 1.8±0.55 | 3.1±0.74 | 2.3±0.58 | 3.9±0.81 | 1.7±0.27 | 3.2±0.69 | 1.8±0.83 | 4.2±1.1 | 0.86±0.25 | n/a | n/a | n/a | n/a |
| | DRAm | 1.5±0.28 | 2.7±0.46 | 2.0±0.44 | 3.5±0.4 | 1.7±0.26 | 2.9±0.34 | 1.4±0.59 | 3.6±0.47 | 0.73±0.05 | n/a | n/a | n/a | n/a |
| | DRA+ | 2.4±0.19 | 4.0±0.46 | 2.9±0.58 | 4.8±0.61 | 1.9±0.21 | 4.1±0.53 | 2.6±0.39 | 5.7±1.6 | 1.2±0.42 | n/a | n/a | n/a | n/a |
| | Non-DRA | 4.1±1.6 | 5.6±1.3 | 3.2±0.9 | 12.7±3.9 | 3.6±1.3 | 5.9±1.0 | 3.2±0.74 | 5.5±2.1 | 6.7±1.6 | n/a | n/a | n/a | n/a |

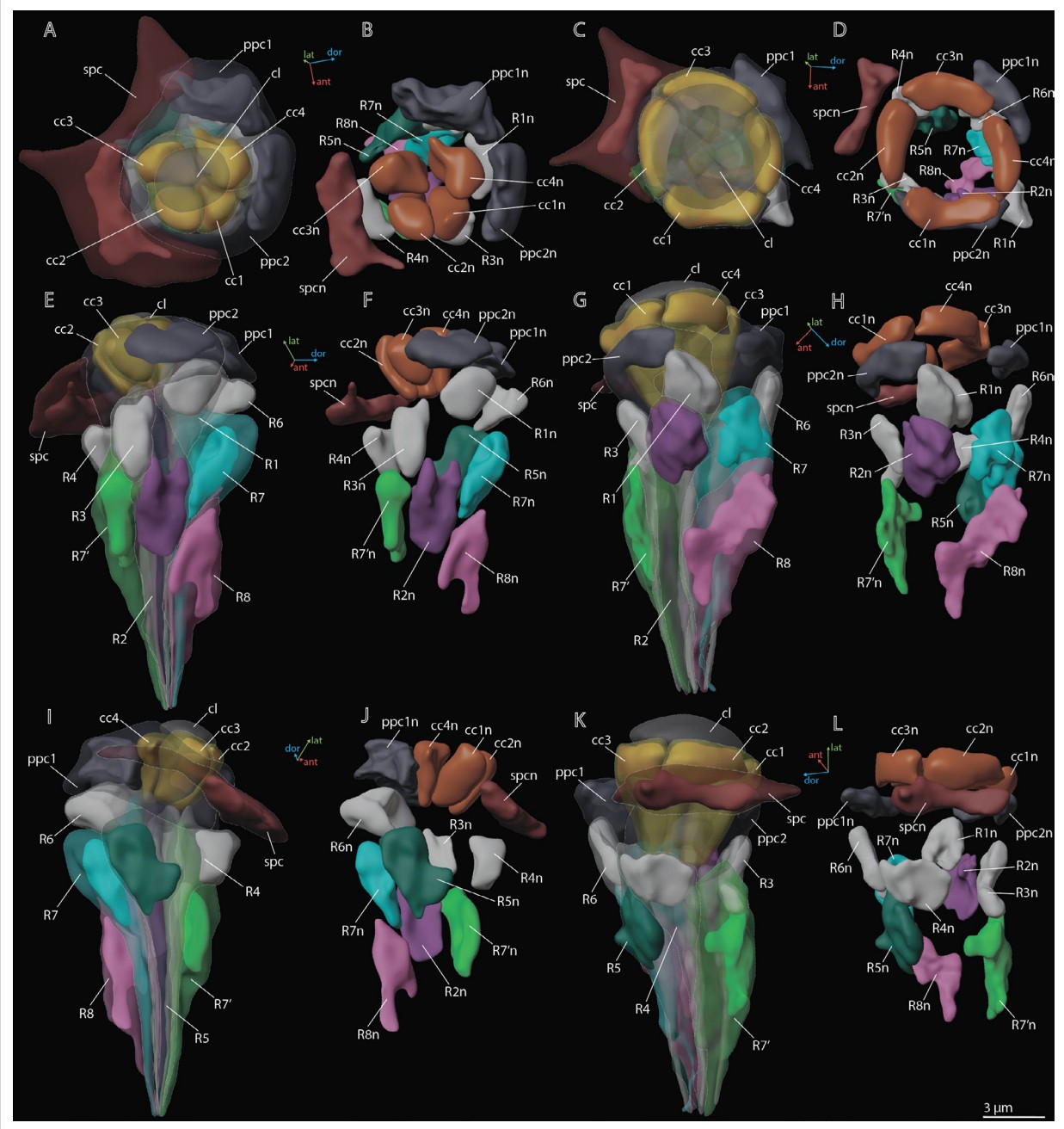

**Figure 4.** 3D reconstruction of nuclei in ommatidium cells of *M. viggianii*. (**A, B, E, F, I, J**) Dorsal rim area (DRA) ommatidia (B6; **C, D, G, H, K, L**) non-DRA ommatidia (C4). cc1–4, crystalline cone cells; cc1n–4 n, nuclei of crystalline cone cells; cl, corneal lens; ppc1, 2, primary pigment cells; ppc1n, ppc2n, nuclei of PPC; R1–R8, retinal cells; R1n–8 n, nuclei of retinal cells; spc, secondary pigment cells; spcn, nuclei of secondary pigment cells. Colors of nuclei same as colors of their cells.

rhabdom length (*Chua et al., 2023*). The length of the rhabdom is nearly equal in DRA and non-DRA ommatidia, 13.4±0.64 µm and 14.2±1.8 µm, respectively (*Table 1*).

The soma of retinula cells is filled with densely packed pigment granules (*Figures 2F and 5*), which are nearly absent in retinula axons. The pigment granules of the retinula cells have an elongated nearly oval shape, with the longest extension parallel to the ommatidium length (*Figures 3 and 5*). The number of pigment granules per cell varies from 70 to 273 and depends on the volume of PR (see *Supplementary file 1b*). The mean volume of a retinula cell pigment granule is 0.049±0.019 µm³.

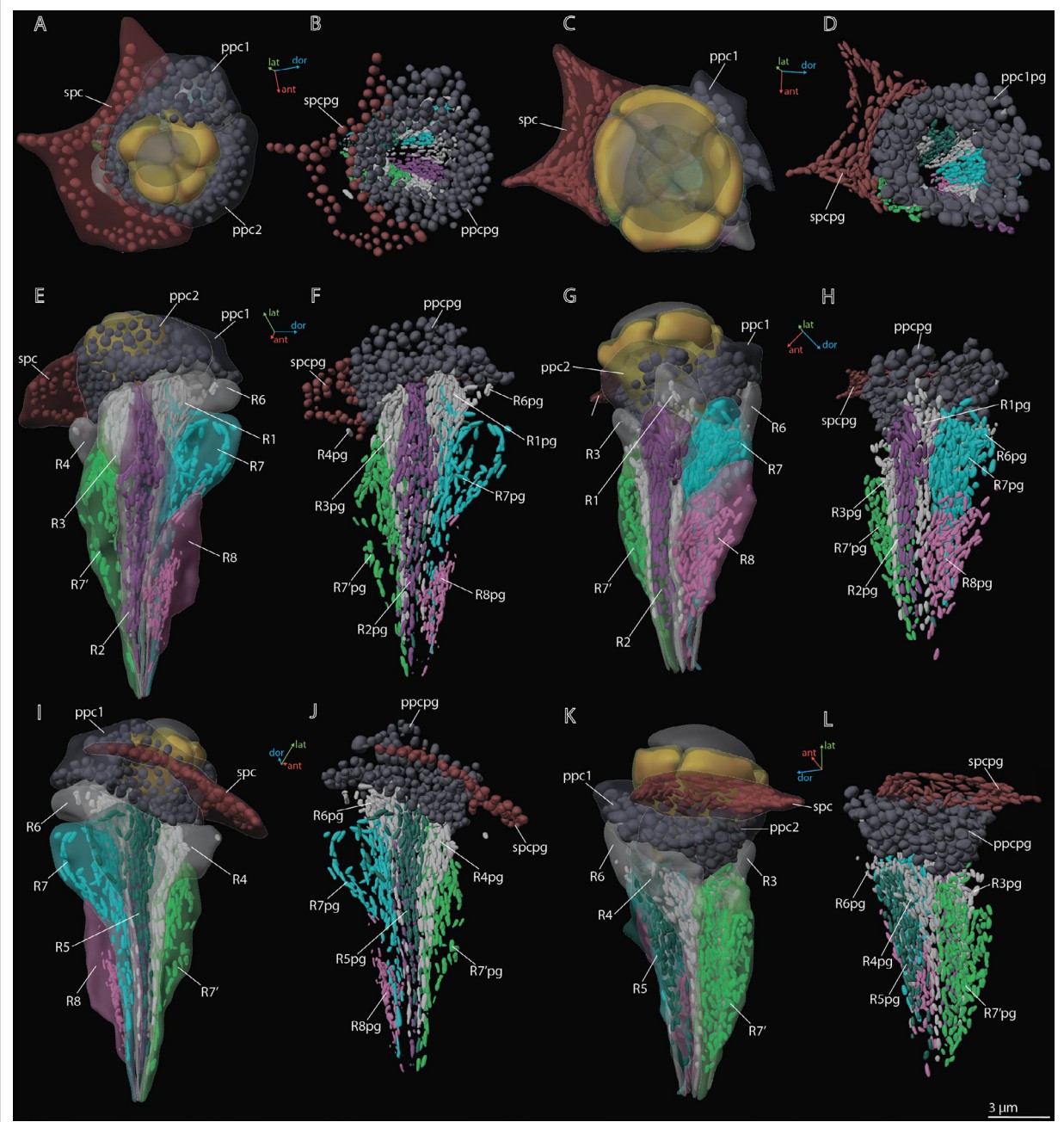

**Figure 5.** 3D reconstruction of pigment granules in the ommatidium cells of *M. viggianii*. (**A, B, E, F, I, J**) Dorsal rim area (DRA) ommatidia (B6); (**C, D, G, H, K, L**) non-DRA ommatidia (C4). ppc1, 2, primary pigment cells; ppc1pg, ppc2pg, pigment granules of PPC; R1–R8, retinal cells; R1pg–R8pg, pigment granules of retinal cells; spc, secondary pigment cells; spcpg, pigment granules of secondary pigment cells. Colors of pigment granules are the same as the colors of their cells.

The distal region of the cells contains more mitochondria profiles than the proximal region (*Figures 3 and 6*; see *Figure 3—figure supplements 1–5*). In retinula cells of DRA ommatidia, the mitochondria are elongated and have numerous units (*Figure 6*). In non-DRA ommatidia, the mitochondria are mostly dendriform. The volume of the chondriome varies from 0.47 μm$^3$ to 4.03 μm$^3$ (see *Supplementary file 1b*). No tracheoles are present in the retina.

## 'Ectopic' photoreceptors (ePR)

We found near the dorsal margin of the eye three 'ectopic' photoreceptor cells (per eye), each of which has several minute rhabdomeres: 8 in ePR1, 9 in ePR2, and 12 in ePR3 (*Figure 7B–D*). These

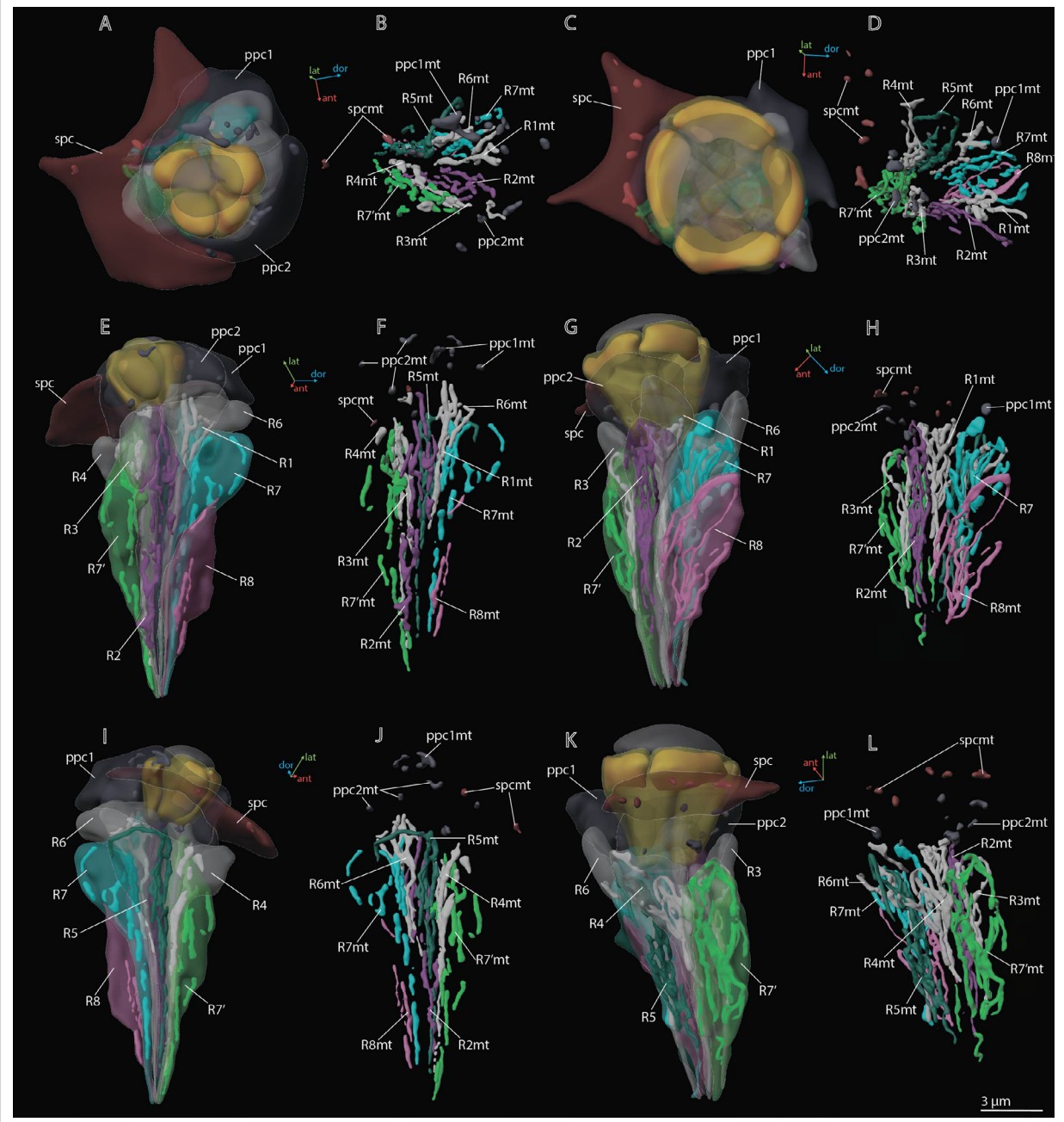

**Figure 6.** 3D reconstruction of mitochondria in the ommatidium cells of *M. viggianii*. (**A, B, E, F, I, J**) Dorsal rim area (DRA) ommatidia (B6); (**C, D, G, H, K, L**) non-DRA ommatidia (C4). ppc1, 2, primary pigment cells; ppc1mt, ppc 2mt, mitochondria of PPC; R1–R8, retinal cells; R1mt–R8mt, mitochondria of retinal cells; spc, secondary pigment cells; spcpg, mitochondria of secondary pigment cells. Colors of mitochondria are the same as the colors of their cells.

cells are situated in the retinal area, behind the first row of DRA ommatidia: ePR1 over D7 and E7, ePR2 over B6, and ePR3 over A5 (*Figure 7A–D*), beneath the cuticle and pigment cells. The ePR have no DA of their own and no connection with the DAs of the adjacent ommatidia. All three cells are drop-shaped and are situated at a distance from each other (they do not touch each other) (see *Video 2*). The mean volume of ePR cell is 25.1±1.4 µm³ (*Table 4*). In the distal part, a large nucleus is situated, its volume being 7.9±0.73 µm³. The mitochondria in ePR are dendriform (*Figure 7B*). The mean volume of the chondriome is 1.9±0.19 µm³. All three cells have their own pigment granules, which resemble in shape and size the pigment granules of retinal cells (*Figure 7B*). The mean volume

**Table 3.** Volumes (μm³) of pigment granules and mitochondria in ommatidia of *M. viggianii* and *Trichogramma evanescens*. The data for *T. evanescens* are from *Fischer et al., 2019*. R1–R8, retinal cells; mt, mitochondria; pg, pigment granules; PPC, primary pigment cells; SPC, secondary pigment cells. B6, C4, A3, A0, ommatidia on which cells pigment granules and mitochondria were reconstructed. Raw data, see *Supplementary file 1c and e.*

| Species | Region | Organelle type | R1 | R2 | R3 | R7' | R4 | R5 | R6 | R7 | R8 | PPC | SPC |
|---|---|---|---|---|---|---|---|---|---|---|---|---|---|
| *M. viggianii* | B6, D7, C4, A3, A0 | PG | 2.5±0.71 | 3.2±0.86 | 2.1±0.51 | 6.5±2.1 | 2.4±0.31 | 2.7±0.76 | 2.2±0.36 | 5.5±2.3 | 2.0±1.0 | 20.6±10.5 | 7.9±3.0 |
| | | Mt | 1.2±0.40 | 1.8±0.56 | 1.0±0.32 | 2.8±0.12 | 1.2±0.39 | 1.8±0.54 | 1.0±0.42 | 2.0±0.59 | 1.5±0.96 | 0.53±0.35 | 0.31±0.070 |
| *T. evanescens* | Three central ommatidia | PG | 2.0±0.12 | 2.1±0.16 | 1.9±0.07 | 2.5±0.19 | 2.0±0.14 | 2.1±0.07 | 1.9±0.15 | 3.0±0.23 | 0.66±0.12 | 8.5±1.4 | 4.0±0.53 |
| | | Mt | 1.8±0.08 | 2.8±0.19 | 1.9±0.05 | 1.4±0.23 | 2.1±0.08 | 2.8±0.14 | 1.9±0.14 | 1.8±0.29 | 0.59±0.05 | 0.46±0.14 | 0.32±0.09 |
| *M. viggianii* | | PG | 12.1±1.9 | 11.4±1.8 | 11.2±2.6 | 13.2±1.7 | 10.5±1.9 | 10.5±1.5 | 10.3±1.5 | 13.2±1.7 | 8.9±3.4 | 32.1±11.5 | 26.1±10.9 |
| | | Mt | 5.7±1.0 | 7.0±1.8 | 4.9±1.3 | 6.6±1.8 | 5.5±1.5 | 6.8±1.0 | 5.0±1.0 | 5.2±0.51 | 5.0±1.9 | 0.60±0.0042 | 1.1±0.42 |
| *T. evanescens* | % | PG | 11.3±1.7 | 7.9±0.63 | 9.6±0.84 | 13.3±2.1 | 11.4±1.7 | 7.9±0.81 | 11.2±0.45 | 11.6±0.43 | 7.8±0.76 | 21.0±0.94 | 19.1±2.4 |
| | | Mt | 9.8±1.1 | 10.3±0.72 | 9.5±0.86 | 7.6±0.51 | 11.6±0.82 | 10.5±0.27 | 11.0±0.24 | 6.9±0.75 | 6.9±0.45 | 1.1±0.17 | 1.5±0.31 |

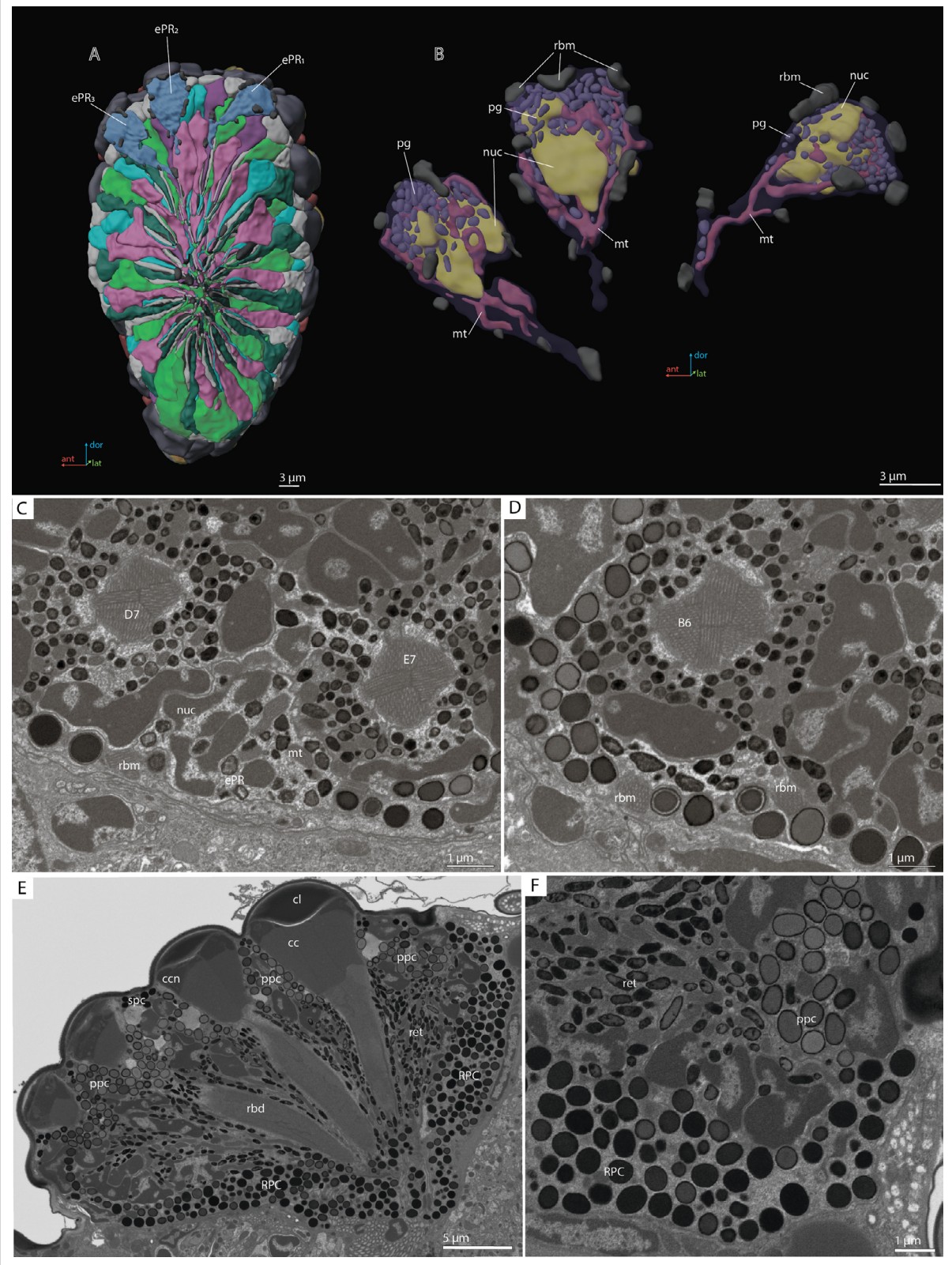

**Figure 7.** 'Ectopic' photoreceptors (ePRs) and rind photoreceptor shield in the compound eye of *M. viggianii*. (**A**) A 3D reconstruction of the eye: posterior view from the retinal area with labeled ePRs; (**B**) a 3D reconstruction of ePRs; (**C, D**) an EM section through dorsal border of the eye; (**F**) an EM section through the eye showing rim pigment cells (RPC). cc, crystalline cones; cl, corneal lens; ePR1-3, 'ectopic' photoreceptors; mt, mitochondria; nuc, nuclei; pg, pigment granules; ppc, primary pigment cells; rbd, rhabdom; rbm, rhabdomeres of 'ectopic' photoreceptors; spc, secondary pigment cells. B6, D7, E7, DRA ommatidia abutting 'ectopic' photoreceptors.

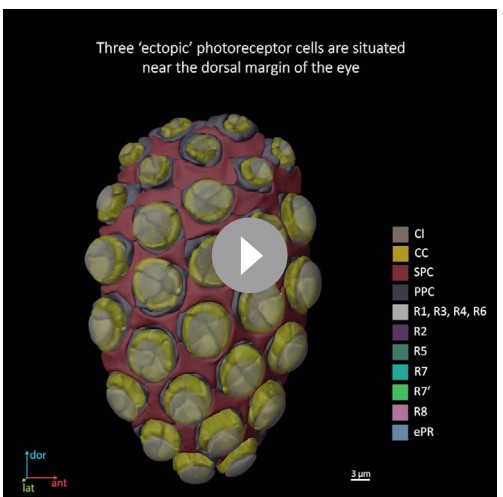

**Video 2.** 3D reconstruction of the ultrastructure of the ePR in the compound eye of *M. viggianii*. cc, crystalline cones; cl, corneal lense; ccn, nuclei of crystalline cone cells; ePR, 'ectopic' photoreceptors; mt, mitochondria; nuc, nuclei; pg, pigment granules; ppc, primary pigment cells; R1–R8, retinal cells; rbm, rhabdomeres; spc, secondary pigment cells.

https://elifesciences.org/articles/103247/figures#video2

of pigment granules is 0.49±0.17 µm³. The axons of the ePR form a bundle and do not project into the lamina, reaching the medulla directly from the eye (*Chua et al., 2023*).

## Rim pigment cells

The eye is surrounded by 16 rim pigment cells (RPC), which are morphologically similar to PPCs (*Figures 2F and 7E, F*). Although they have not been reconstructed because identifying their boundaries is difficult, their number was determined by counting the nuclei surrounding the eye. RPCs are filled with pigment granules of spherical shape. The mean volume of one pigment granule is 0.18±0.049 µm³.

## Discussion

The general structure of the compound eye in *M. viggianii* is similar to that previously described in *M. polilovi*, misidentified earlier as *M. mymaripenne* (*Makarova et al., 2015*) and subsequently described as a new species (*Polaszek et al., 2022*). In contrast to the oblong and 'narrow' ommatidia in the eyes of other minute hymenopterans (*Fischer et al., 2011*; *Makarova et al., 2015*; *Fischer et al., 2019*), the ommatidia in *M. viggianii* are short and 'wide'.

## DRA and non-DRA ommatidia

The results of the 3D reconstruction, morphometry, and volumetry of the key components of the eye and data on the connectome of the lamina *Chua et al., 2023* have shown considerable differences (corneal and retinal) between DRA and non-DRA ommatidia (*Table 5*; see *Video 3*). Morphometric analysis clearly reveals a group of seven ommatidia (D7, E7, B6, C6, D6, A5, and B5) in the dorsal area of the eye (DRAm) (*Tables 1 and 2*; *Figure 2A*). The analysis of synaptic connections of R7 and R7' supplements this group with three more ommatidia (E6, C5, and A4) (DRA(+)) (*Chua et al., 2023*; *Figure 2A*; see *Supplementary file 1a and b*). According to their morphological characters, the ommatidia DRA+ have features of both DRAm ommatidia and non-DRA ommatidia. Judging by the size of the cone and lens and by the position of the nuclei in the cells of the cone, ommatidium E6 is more similar to DRAm ommatidia (*Figure 2*; see *Supplementary file 1a and b*). The nuclei occupy almost the entire cell volume in DRAm ommatidia. In two other ommatidia, DRA+ (C5 and A4), the DA is more similar to that of non-DRA ommatidia (the nuclei of the cone cells form an aperture under the lens and are situated in the dorsal third of the cells). The volumetric parameters of DRA+ommatidia

**Table 4.** Volumes (µm³) and number for 'ectopic' photoreceptors in *M. viggianii*.
EPR1–3, 'ectopic' photoreceptors.

| | Soma | Nuclei | Rhabdomeres (total volume) | Pigment granules (total volume) | Number of pigment granules per cell | Mitochondria total volume | Number of mitochondria per cell |
|---|---|---|---|---|---|---|---|
| EPR1 | 26.5 | 8.8 | 2.5 | 2.8 | 117 | 1.8 | 5 |
| EPR2 | 25.1 | 7,5 | 2.4 | 3.2 | 130 | 2.2 | 5 |
| EPR3 | 23.6 | 7.6 | 2.2 | 3 | 105 | 1.9 | 10 |
| Mean | 25.1±1.4 | 7.9±0.73 | 2.4±0.15 | 2.9±0.22 | 117±12 | 1.9±0.19 | 7±3 |

**Table 5.** Features of DRA and non-DRA ommatidia obtained by reconstruction of compound eyes of *M. viggianii*. DRA, dorsal rim ommatidia in general (DRAm and DRA+); DA, dioptric apparatus; non-DRA, regular (non-DRA) ommatidia; PC, pigment cells; PPC, primary pigment cells; R1–R8, retinal cells. *t*-test *0.001≤p<0.01, **0.0001≤p<0.001, ***p<0.0001.

| Ommatidial area | Features | | DRA | Non-DRA | Difference |
|---|---|---|---|---|---|
| | Ommatidial length, μm | | 19.2±0.37 | 22.3±2.5 | ** |
| | Cone | Length, μm | 3.0±0.37 | 4.6±0.59 | *** |
| | | Width, μm | 4.5±0.21 | 6.6±0.25 | *** |
| | | Volume, μm³ | 13.0±6.0 | 32.9±5.9 | *** |
| | Lens | Lens diameter, μm | 6.9±0.89 | 8.0±0.77 | * |
| | | Volume, μm³ | 16.5±14.0 | 41.4±9.0 | ** |
| | | Lens thickness, μm | 2.5±0.56 | 3.3±0.39 | * |
| | | Inner curvature, μm | 1.2±0.32 | 3.0±0.51 | *** |
| | | Outer curvature, μm | 3.3±0.63 | 4.7±0.39 | *** |
| | Cone cell (CC) nuclei | CC nuclei | Fill most of the volume of the cell | Form a ring in the upper third of the cone | n/a |
| | | % of cone volume | 58±0.17 | 23±0.02 | *** |
| DA | | Volume, μm³ | 6.6±0.68 | 7.5±1.1 | *** |
| PC | PPC volume | | 47.1±14.3 | 64.5±11.4 | *** |
| | Rhabdom | Shape | Rectangular shape of rhabdom from center to lower part | Spheric shape of the rhabdom along whole length | n/a |
| | | Diameter, μm | 2.0±0.16 | 2.7±0.75 | *** |
| | Microvilli orientation | | Orthogonal orientation of R7, R7' microvilli along the rhabdom length | Non-orthogonal orientation of R7, R7' | n/a |
| | Volume, μm³ | Duet (R2, R5) | 24.8±3.9 | 30.3±3.2 | *** |
| | | Quartet (R1, R3, R4, R6) | 17.8±1.9 | 21.4±3.5 | *** |
| | | R7' | 33.0±2.5 | 53.3±11.1 | *** |
| | | R7, R7' | 32.6±3.7 | 41.9±15.1 | *** |
| Retina | | R8 | 15.8±1.9 | 28.2±4.2 | *** |

are slightly greater than those of DRAm ommatidia but smaller than those of non-DRA ommatidia (*Table 2*). The morphology and the retinotopic pattern of ommatidial specialization in the eye suggest that DRA+ommatidia lie in the transitional zone between specialized and non-specialized ommatidia.

Specialized ommatidia of DRAs in the compound eyes were described in many insects (Odonata, Orthoptera, Hemiptera, Coleoptera, Hymenoptera, Lepidoptera, Diptera, and others; summarized in *Labhart and Meyer, 1999*; *Labhart et al., 2009*). The results of our morphological analysis of all ommatidia in *Megaphragma* are consistent with the light-polarization related features in Hymenoptera and other insects (e.g., *Gribakin, 1972*; *Menzel and Snyder, 1974*; *Schinz, 1975*; *Labhart, 1980*; *Meyer and Labhart, 1981*; *Aepli et al., 1985*; *Menzel et al., 1991*; *Labhart and Meyer, 1999*; *Wehner and Labhart, 2006*; *Greiner et al., 2007*; *Narendra et al., 2013*; *Jie et al., 2023*). Moreover, it agrees well with the regional specialization of DRA ommatidia manifested in the orientation of microvilli and synaptic connectivity in lamina cartridges (*Chua et al., 2023*).

## Corneal specializations

There is a significant difference in the length and volume of DA between DRA and non-DRA ommatidia (*Tables 1, 2 and 5*). The lenses in DRA ommatidia are visually different from those in non-DRA ommatidia (*Figures 2 and 4*). Considerable differences are visible in the diameter, thickness of the cornea,

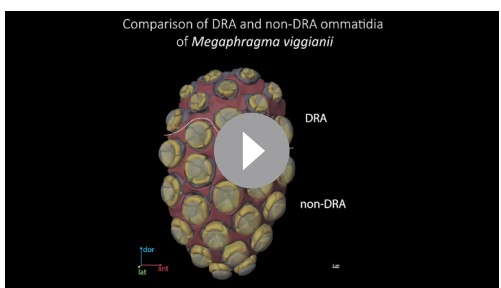

**Video 3.** 3D reconstruction of the ultrastructure of the DRA and Reg ommatidia in *M. viggianii*. DRAm, dorsal rim area ommatidia (morphological specialization); DRA+, transitional zone ommatidia; cc, crystalline cones; cl, corneal lense; ccn, nuclei of crystalline cone cells; ppc, primary pigment cells; R1–R8, retinal cells; rbm, rhabdomeres; spc, secondary pigment cells. Colors of nuclei, mitochondria, and pigment granules are the same as the colors of their cells.

https://elifesciences.org/articles/103247/figures#video3

radius of curvature (*Figure 8D*), and volume of the lenses, which are smaller in DRA ommatidia (see *Supplementary file 1d*). Differences are found also in the calculated focal lengths (*Figure 8E*), which are smaller in DRA (and intermediate in E6, which has a similar cone structure) than in non-DRA ommatidia. The volume of the lenses and of the cone cells also supports the division into the areas DRAm and DRA+ (*Table 2*; *Figure 8F*).

DRA ommatidia are characterized by smaller DA, lenses, and cone cells than non-DRA ommatidia (*Tables 1 and 2*). The nuclei of the cone cells of DRAm ommatidia (and those of ommatidium E6 of DRA+) occupy almost the entire cell volume. The chromatin of the nuclei is strongly compacted and occupies almost all of the volume of each nucleus (*Figure 3*). Since the cone cells are adjacent to each other over their entire length, their nuclei form an electron-dense formation under the lens (*Figures 3 and 4*; see *Figure 3—figure supplements 1–5*, *Videos 1 and 3*). Since the nuclei in DRA and non-DRA ommatidia are arranged differently in cone cells, we suggest that the nuclei of the cone cells of DRA ommatidia in *M. viggianii* perform some optical role in facilitating the specialization of this group of ommatidia. The optical function for nuclei was described for rod cells of nocturnal vertebrates, where the chromatin inside the cell nucleus has a direct effect on light propagation (*Solovei et al., 2009*; *Błaszczak et al., 2014*; *Feodorova et al., 2020*).

## Retinal specializations

The group of short visual fiber PRs (R1–R6) is clearly divided into the duet (R2, R5) and quartet (R1, R3, R4, R6), according to the volume of the cells, which agrees with the data on lamina circuits (*Chua et al., 2023*) and other data (*Friedrich et al., 2011*). Volumetric analysis of PR subtypes among all ommatidia and between DRA and non-DRA groups shows significant differences (*Table 5*). The Kruskal–Wallis test for all ommatidia shows H (3, N=261)=146.3, p<0.0001; H (3, N=90)=69.3 б p=0.0000 for DRA and H (3, N=171)=106.1, p=0.000 for non-DRA. The smaller volume among the short PRs belongs to the quartet (17.8±1.9 µm³ and 21.4±3.5 µm³ for DRA and non-DRA ommatidia, respectively). The duet PR has a mean volume of 24.8±3.9 µm³ in DRA and 30.3±3.2 µm³ in non-DRA ommatidia. The greater volume among PRs belongs to R7' cells and is 33.0±2.5 µm³ in DRA, and 53.3±11.1 µm³ in non-DRA ommatidia. The basal PR cell, R8, has a volume of 15.8±1.9 µm³ in DRA and 28.2±4.2 µm³ in non-DRA ommatidia.

Rhabdom volume gradually increases from the dorsal to the ventral area of the eye (*Figure 8A*; see *Supplementary file 1b*). The volumes of the rhabdomeres of the photoreceptor in all ommatidia of the eye vary (*Table 2*) and reveal a general trend for an increase with total cell volume. The rhabdomeres of cells R7' and R8 are much smaller in DRA ommatidia than in non-DRA ommatidia (*Figure 8B and C*; *Table 2*). Opposing twin rhabdomeres R7 and R7' do not show any significant differences in volumes (*Table 2*), but orthogonal orientations of their microvilli support the role played by DRA in detecting light polarization (*Chua et al., 2023*). Analysis of the orientation of the microvilli of the rhabdom has shown that in DRA ommatidia rhabdomeres of the R7 and R7' cells are consistent throughout the depth of the ommatidium and orthogonal to each other, which is consistent with the disparity in the numbers of synapses received by R7 and R7' in the lamina (*Chua et al., 2023*).

In spite of the difference in length between DRA and non-DRA ommatidia, the length of the rhabdoms in the ommatidia differs little: 13.4±0.64 µm for DRA and 14.2±1.8 µm for non-DRA, which is much smaller than in *Apis mellifera* (200–500 µm) (*Menzel et al., 1991*), and comparable with that of *Trichogramma evanescens* (18.3±0.28 µm) (*Fischer et al., 2019*). The total rhabdom shortening in *M. viggianii* ommatidia probably favors polarization and absolute sensitivity by reducing self-screening

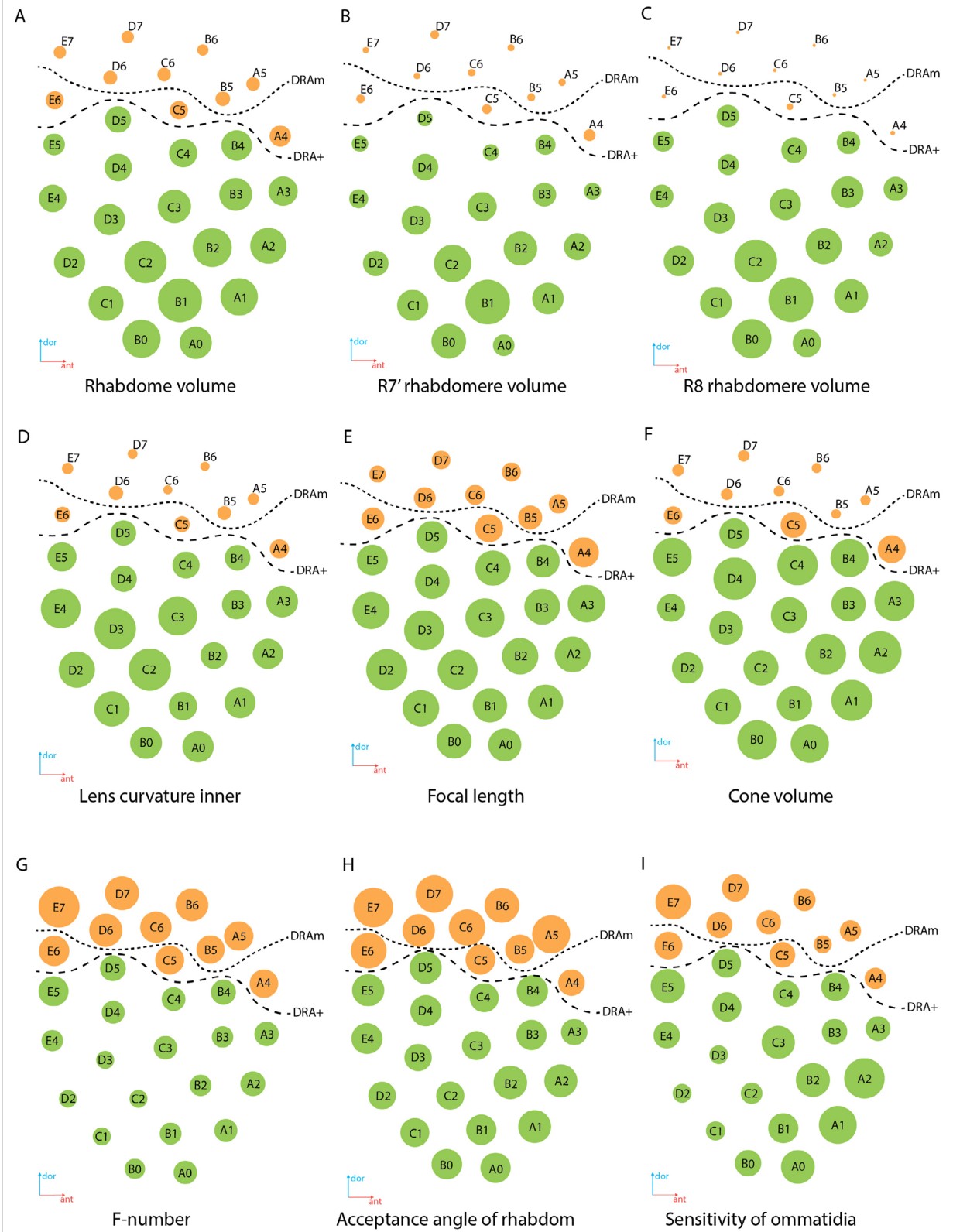

**Figure 8.** Regional specialization of the compound eye in *M. viggianii*. Bubble size indicates the value of each parameter. (**A**) Rhabdom volume; (**B**) volume of rhabdomere R7'; (**C**) volume of rhabdomere R8; (**D**) inner curvature of the lens; (**E**) focal length; (**F**) cone volume; (**G**) f-number; (**H**) acceptance angle of the rhabdom; (**I**) sensitivity of the ommatidium. DRAm, dorsal rim area ommatidia (morphological specialization); DRA+, transitional zone ommatidia. Ommatidia are named as in *Chua et al., 2023*.

and widening the rhabdomeric cross-sectional area (*Nilsson et al., 1987*; *Labhart and Meyer, 1999*; *Wehner and Labhart, 2006*).

The cross-sections of the distal rhabdom in DRA and non-DRA ommatidia differ. Rhabdoms in DRA possess partly rectangular cross-sectional profiles (*Figure 3*; see *Figure 3—figure supplements 1–5*), similar to the profiles of polarization of sensitive ommatidia in the DRA of ants and bees (*Gribakin, 1972*; *Menzel et al., 1991*; *Labhart and Meyer, 1999*; *Greiner et al., 2007*). In the DRA ommatidia of *M. viggianii,* the rhabdom cross-sectional shape is not constant throughout its length. In the first half (under the lens) the shape of the rhabdom is nearly round and becomes rectangular or nearly square in the center (*Figure 3D–F*; see *Figure 3—figure supplements 1–5*, *Figure 2* ommatidia). Non-DRA ommatidia possess round profiles over the entire length of the rhabdom (*Figure 3J–L*; see *Figure 3— figure supplements 1–5*; *Figure 2* of non-DRA ommatidia).

In DRA ommatidia, rhabdoms have smaller volumes and narrower distal parts than in non-DRA ommatidia (see *Supplementary file 1b*; *Figure 3—figure supplements 1–5*).

## Optical properties of DRA

Some optical parameters differ between ommatidia within the eye (*Figure 8E, G–I*; see *Supplementary file 1d*). The short focal length in DRA (*Figure 8E*) in combination with rhabdom diameters results in the relatively large acceptance angles of the rhabdoms (*Figure 8H*). The estimated optical sensitivity of the eyes is very close to those reported for diurnal hymenopterans with apposition eyes (*Greiner et al., 2004*; *Gutiérrez et al., 2024*) and possess around $0.19 \pm 0.04$ μm$^2$ sr (*Figure 8I*). *M. viggianii* have large values of acceptance angle $\Delta \rho$ , and thus should result in a low spatial resolution (see *Supplementary file 1d*).

## Other findings
### Retinula cells and rhabdom

Despite the extreme miniaturization of the eye leading to the dense packing of ommatidia components and lack of space, traces of structural diversification of PRs are retained and indicate a strong evolutionary conservation. Morphological differences of photoreceptors as a key to spectral sensitivity cells were suggested by *Gribakin, 1975*. The division into the duet and quartet PRs is an ancestral trait of the insect retina (*Friedrich et al., 2011*). The position of the nuclei of the outer PR quartet (R1, R3, R4, and R6) is different from that of the duet (R2 and R5) and from that reported for bees (*Gribakin, 1975*), ants (*Herrling, 1976*), beetles (*Schmitt et al., 1982*), dragonflies (*Meinertzhagen et al., 1983*), and butterflies (*Awata et al., 2010*). The most distal position of the nuclei of the outer PR quartet relative to the duet is also found in *Megaphragma* (*Figure 4F, H, J and L*). Such differences between cells are an indication of the strong evolutionary conservation of the outer PR quartet and duet subgroups and can be attributed to their wavelength sensitivities (*Friedrich et al., 2011*).

In *Megaphragma,* the PRs duet projects deeper into the lamina than the quartet PRs, as in most Hymenoptera (*Ribi, 1975*; *Greiner et al., 2004*). Duet and quartet PR also have different arrangements in axon bundles and numbers of output synapses in the lamina: duet synapses are distributed over the whole length of the lamina cartridge, but quartet synapses are distributed in the anterior part of the cartridge (*Chua et al., 2023*). It has been suggested that the outer PR duet provides information for motion-detecting vision, while the quartet PRs participate in color vision (*Takemura and Arikawa, 2006*).

Morphological analysis of the ommatidia showed that in most ommatidia the cytoplasm and the rhabdomere of R7 are distinguished by a lower electron density than adjacent cells (see *Figure 3— figure supplements 1–5*). This may be the result of accidental bleaching of the long wave receptors during fixation of the sample and of the reaction of $OsO_4$ as a 'developer' of light-induced changes in cells (*Gribakin, 1975*). Retinula cells R7 and R7' have been reported as UV sensors in hymenopteran ommatidia (*Wakakuwa et al., 2007*; *Spaethe and Briscoe, 2005*). This can indirectly explain the relatively larger volume of R7' cells and rhabdomeres, and the highest pigment granule volumes in R7 and R7' compared with other retinula cells in the eye of *Megaphragma*. This agrees well with the connectivity pattern findings in the lamina (*Chua et al., 2023*).

The volume and number of pigment granules and mitochondria per cell positively correlate with the volume of PR (*Tables 2 and 3*). The presence of numerous mitochondrial profiles visible in most single sections in the distal part of ommatidia is a result of the sectioning of few dendriform-like units,

rather in non-DRA than in DRA ommatidia (*Figures 3 and 5*). The distal parts of the cells are referred to as the most active metabolically, thus indicating an exponential gradient of light absorption (*Gribakin, 1975*).

## Pigment cells

The volume of PPC and SPC is greater in DRA ommatidia than in non-DRA ommatidia (*Table 2*). The shapes and volumes of pigment granules differ in PPC, SPC, and PR (*Figures 2F, 3, and 7E, F*). The pigment granules of all cell types also vary in electron density. PPC have lower electron density than SPC and granules of PR (*Figure 6*). This difference in electron density could be an indication of different biochemical activity and shielding functions of light-absorbing pigment granules (*Gribakin, 1981*).

The total volume of pigment granules is higher in pigment cells than in photoreceptor cells. Pigment granules occupy about 32% of PPC and 26% of SPC volume. By contrast, pigment granules of the retinal cells occupy 10–13% of cell volume.

A total of 24 SPCs were identified in the whole eye of *M. viggianii*. However, we cannot clearly identify them with any particular ommatidium because they probably perform their screening function for adjacent ommatidia (*Figure 2A and F*). Pigment granules of SPC vary in shape in different parts of the eye. Preliminary observations show that the SPCs near the DRA ommatidia have round pigment granules (*Figure 5B*). The SPCs that surround the area of non-DRA ommatidia in the center and proximal third of the eye have more oval pigment granules (*Figure 5D*).

Sixteen RPC ensheathe the eye laterally and prevent light from passing from areas outside the compound eyes onto the photoreceptors (*Stavenga and Hardie, 1989*; *Stavenga, 2002*; *Tomlinson, 2012*; *Mohr and Fischer, 2020*). In spite of the same volume and shape of granules in PPC and RPC, the RPC pigment granules have a high electron density, comparable to SPC (*Figure 7E and F*). Studies on *Drosophila* have shown that the pigment rim originates from secondary/tertiary-like pigment cells of the pupa (*Wolff and Ready, 1991*).

## Comparison with *Trichogramma*

The complete cellular-level 3D reconstruction of the entire eye of one of the smallest insects provides the most detailed information about the structure of the insect compound eyes in general. However, the uniqueness of the dataset complicates an extensive comparative assessment of the results, in particular the volumetric ones. There exists a pioneering 3D reconstruction of three ommatidia from the eye of a male *Trichogramma evanescens* by *Fischer et al., 2019*. Considering that the data for *T. evanescens* was obtained from the central ommatidia (*Fischer et al., 2019*), we can tentatively assume that they were non-DRA.

In most aspects, the eyes of *Megaphragma* are smaller than those of *Trichogramma* and contain at least four times fewer facets (*Makarova et al., 2015*). But individual facets of *Megaphragma* eyes have a wider lens diameter than those of the eyes of *Trichogramma*. The small number of ommatidia in *Megaphragma* in comparison to *Trichogramma* is probably compensated for by the larger diameter of the facets, wider and shorter rhabdoms, and short DA (*Makarova et al., 2015*).

The location of the nuclei in the duet (R2, R5) and R7 equivalent cells (R7, R7') differ in *Trichogramma* and *Megaphragma*. In *T. evanescens*, the nuclei of the duet occupy a proximal position approximately half-way along the ommatidia, and the nuclei of R7 and R7' cells are shifted ventrally to the level slightly above that for the duet (*Fischer et al., 2019*). In *Megaphragma*, the nuclei of the duet are similar or more distal compared to that in R7 and R7' (*Figure 4*).

## Number of SPCs

The main structural difference between *Megaphragma* and *Trichogramma* ommatidia is the number of SPCs. According to the description of *Trichogramma* ommatidia, six (*Fischer et al., 2010*) or five (*Fischer et al., 2019*) SPCs are positioned directly beneath the cornea and envelop PPCs in their dorsal third (*Fischer et al., 2019*). There is no information about the total number of SPCs in the eye of *Trichogramma* or any other insect. In the right eye of *Megaphragma*, there is a total of 24 SPCs. According to 3D reconstructions, only the PPCs of the central rows of ommatidia can be 'encircled' by four SPC; the outer rows are abutted by two or three SPCs. The reduction of the number of SPC could be a result of miniaturization in *Megaphragma* eyes.

## Mitochondria

The second structural difference is the absence in the cone cells of *Megaphragma* ommatidia of mitochondria, which are present in *Trichogramma* (*Fischer et al., 2019*). Although the cones of *Megaphragma* do not contain bona fide mitochondria, we found electron-dense elements, which could be residual bodies, near the border of the cone cells, where the mitochondria of *Trichogramma* were reported (*Figure 3G and I*; see *Figure 3—figure supplements 1–5* (A3, C4, B3)).

There are also differences in the shape and number of mitochondria in the retinal cells. In *Trichogramma,* there are elongated mitochondria in the PR cells, but in *Megaphragma* most of the mitochondria are dendriform.

Finally, the total volume of mitochondria in retinula cells is higher in *Trichogramma* than in *Megaphragma* (*Table 3*).

## Pigment granules

Despite the smaller volume of cells, *Trichogramma* has a higher number of pigment granules in PPC/SPC ($212\pm50/255\pm16$) (*Fischer et al., 2019*) than *Megaphragma* (about $158\pm29/144\pm52$). The mean volume of individual pigment granules differs between the two genera. The mean unit volume in *Trichogramma* SPC is $0.017\pm0.03$ $\mu m^3$ (*Fischer et al., 2019*) and $0.050\pm0.016$ $\mu m^3$ in *Megaphragma*. The mean unit volume in PPC granules is greater in *Trichogramma* ($0.52\pm0.1$ $\mu m^3$) (*Fischer et al., 2019*) than in *Megaphragma* ($0.18\pm0.039$ $\mu m^3$). The total volume of pigment granules of pigment cells is greater in *Megaphragma* than in *Trichogramma* (*Table 3*). However, the measurements of Fischer and coauthors contain some discrepancies in the values of the general volumes of pigment granules,

**Table 6.** Comparison of volumes ($\mu m^3$) of ommatidial components for *Trichogramma evanescens* and *M. viggianii*.
Data on *T. evanescens* are from *Fischer et al., 2019*. CC, crystalline cones R1–R8, retinal cells; PPC, primary pigment cells; SPC, secondary pigment cells.

| | CC | PPC | SPC | R1 | R2 | R3 | R4 | R5 | R6 | R7' | R7 | R8 |
|---|---|---|---|---|---|---|---|---|---|---|---|---|
| **Soma** | | | | | | | | | | | | |
| *T. evanescens* | 12.3±1.2 | 40.5±6.9 | 21.0±2.5 | 18.5±2.3 | 27.6±0.35 | 20.3±1.3 | 18.2±1.4 | 26.9±1.8 | 17.5±0.92 | 19.7±4.3 | 26.5±2.8 | 8.48±1.1 |
| *M. viggianii* (B3, C3, C4) | 32.9±7.5 | 61.8±9.2 | 23.2±3.8 | 20.4±0.81 | 30.3±3.0 | 19.4±1.8 | 21.2±1.6 | 30.2±2.0 | 18.1±1.7 | 48.4±8.4 | 33.8±10.9 | 26.9±1.7 |
| *M. viggianii* (all non-DRA) | 32.9±5.9 | 64.4±11.4 | 24.5±3.1 | 20.9±3.2 | 29.4±3.1 | 22.0±2.6 | 22.3±3.5 | 31.1±3.2 | 20.4±4.5 | 53.3±11.1 | 30.6±8.5 | 28.1±4.3 |
| **Nuclei** | | | | | | | | | | | | |
| *T. evanescens* | 3.1±0.18 | 3.6±0.29 | 3.2±0.40 | 2.4±0.09 | 2.4±0.15 | 2.4±0.17 | 2.1±0.11 | 2.2±0.24 | 2.3±0.1 | 2.4±0.3 | 2.9±0.48 | 1.9±0.15 |
| *M. viggianii* (B3, C3, C4) | 7.0±1.0 | 7.2±1.1 | 5.7±0.28 | 7.0±0.11 | 7.9±0.18 | 6.7±0.63 | 6.9±0.65 | 8.0±0.48 | 5.9±0.36 | 9.1±1.1 | 8.1±1.3 | 8.3±0.73 |
| *M. viggianii* (all non-DRA) | 7.5±1.1 | 7.6±0.98 | 6.4±1.03 | 7.0±0.56 | 7.6±0.90 | 7.2±0.72 | 7.2±0.84 | 7.9±0.71 | 6.5±1.7 | 9.2±1.0 | 7.9±1.2 | 8.3±0.80 |
| **Rhabdomere** | | | | | | | | | | | | |
| *T. evanescens* | n/a | n/a | n/a | 2.4±0.2 | 3.5±0.19 | 2.2±0.02 | 2.3±0.08 | 3.5±0.15 | 2.4±0.25 | 2.1±0.27 | 3.8±0.28 | 0.79±0.07 |
| *M. viggianii* (B3, C3, C4) | n/a | n/a | n/a | 3.7±0.92 | 6.0±0.79 | 2.3±1.0 | 3.2±0.46 | 5.9±0.54 | 2.8±0.071 | 11.1±3.0 | 6.5±2.3 | 6.6±0.80 |
| *M. viggianii* (all non-DRA) | n/a | n/a | n/a | 4.1±1.6 | 5.6±1.3 | 3.2±0.94 | 3.6±1.3 | 5.9±0.97 | 3.2±0.74 | 12.7±3.9 | 5.5±2.1 | 6.7±1.6 |
| **Nuclei/soma, %** | | | | | | | | | | | | |
| *T. evanescens* | 25.0±2.7 | 9.1±1.2 | 15.2±2.5 | 13.0±1.1 | 8.8±0.55 | 11.6±0.67 | 12.4±1.6 | 12.0±0.41 | 8.1±0.44 | 13.3±1.02 | 11.0±1.02 | 23.0±1.3 |
| *M. viggianii* (B3, C3, C4) | 22.5±7.1 | 11.7±1.7 | 25.0±4.5 | 34.4±1.7 | 26.5±3.3 | 34.7±1.0 | 19.0±0.98 | 32.4±1.2 | 26.6±0.33 | 32.9±1.3 | 25.1±6.8 | 30.8±0.74 |
| *M. viggianii* (all non-DRA) | 35.3±19.9 | 15.1±11.6 | 28.6±7.9 | 34.5±3.4 | 27.1±2.9 | 34.4±3.1 | 20.1±4.2 | 34.2±3.5 | 27.0±3.1 | 33.5±6.2 | 26.7±4.6 | 34.8±7.8 |

which cannot be so small given the number and diameter of the pigment granules of PPC (*Fischer et al., 2019*).

The number of pigment granules in the retinal cells of *Megaphragma* (from 70 to 270 and depending on the cell size and ommatidia type [Dra of non-DRA]) is higher than in *Trichogramma* (40–80, depending on cell size) (*Fischer et al., 2019*; see *Supplementary file 1b*). Despite this, the total volume of pigment granules per cell is close in *Megaphragma* and *Trichogramma* (*Table 3*).

Despite the similar volume of the R7 cells, the total volume of pigment granules in the R7 cells in *Megaphragma* is almost twice as great as in *Trichogramma* (*Table 3*). As in *Trichogramma*, the R7′ and R7 cells in *Megaphragma* display a higher pigment granule volume in comparison to those of other photoreceptors, which could indirectly implicate them as UV sensors (*Wakakuwa et al., 2007*; *Spaethe and Briscoe, 2005*). But in *Megaphragma* R7 does not have a high rhabdomere volume (whereas R7′ does have a high volume), in contrast to *Trichogramma*, in which R7′ does not stand out among other retinula cells (*Table 6*). This difference could be a result of the functional diversification between R7 and R7′ in different species or expression of different opsin paralogs in the same ommatidia (*Friedrich et al., 2011*).

## Volumetry

In addition to revealing morphological differences of ommatidia, we compared their volumes in two ways: we compared the three ommatidia studied in *Trichogramma* with three central ommatidia of *Megaphragma* (B3, C3, C4) and with all non-DRA ommatidia of *Megaphragma* (*Table 6*).

Despite having smaller eyes and body lengths (~290 µm) and fewer ommatidia, *M. viggianii* cells have 0.9–3.3 times the volume found in *T. evanescens,* which has a body length of 400–500 µm (*Table 6*). The most prominent difference in volume was revealed between the R8 PRs (more than 3 times greater in *Megaphragma* than in *Trichogramma*), the R7′ PRs (2.5 times larger in *Megaphragma*), and the cone cells (2.5 times greater in *Megaphragma*). The minor difference between the majority of retinal cells can be explained as a result of the slimmer ommatidia of *Trichogramma* eyes or by the sex differences: *Trichogramma* males have smaller eyes and shorter ommatidia than females (*Fischer et al., 2011*; *Fischer et al., 2019*).

The volumes of the nuclei in all cells in *Megaphragma* ommatidia are 1.7–4.2 times greater than in *Trichogramma* photoreceptors (*Table 6*) and interneurons (*Fischer et al., 2018*), despite the smaller difference in body size. The percentages for cell volume (soma) occupied by the nuclei in non-DRA ommatidia of *Megaphragma* are higher than in *Trichogramma*, constituting up to 23% in cone cells, 12% in PPC, 25% in SPC, and 19–34% in PR (*Table 6*; *Supplementary file 1c*). The mean nucleus volume in the ommatidium cells of *Megaphragma* is also similar to the nucleus volume of most Johnston's organ cells (*Diakova et al., 2022*). *Megaphragma* nuclei are also characterized by a more compacted chromatin. The greater volume of nuclei in spite of smaller eyes can be explained by the fact that *M. viggianii* has one of the largest genome sizes in Chalcidoidea (*Sharko et al., 2019*).

## 'Ectopic' photoreceptors (ePR)

We have revealed photoreceptor cells that are not connected with the DA of the eye (*Figure 7*). The presence of such cells is confirmed in stacks of three heads of *Megaphragma* (two females and one male). The number of cells in all samples is invariably three. Tracing the projections of these *ePR* demonstrates that their axons form a bundle, do not project into the lamina, and reach the medulla directly from the eye. In the region of the lamina, they squeeze between two cartridges before projecting the medulla (*Chua et al., 2023*). Their morphology is closer to those of R8 than to those of any other cell type. Although these ePR axons lack corresponding LMCs, they exhibit similar ramification and projection into the medulla as R8, and form connections with cells that synapse with R8 in other medulla columns (*Chua et al., 2023*). Having no cone or lens, their small rhabdomeres may receive unfocused light. This could potentially be used to measure ambient light intensity and may be helpful in regulating circadian rhythms (*Chua et al., 2023*). The position of ePR, their morphology, and synaptic targets look similar to the eyelet (extraretinal photoreceptor cluster) discovered in *Drosophila* (*Helfrich-Förster et al., 2002*). Eyelets are remnants of the larval photoreceptors, Bolwig's organs in *Drosophila* (*Hofbauer and Buchner, 1989*). Unlike *Drosophila*, Trichogrammatidae are egg parasitoids and their central nervous system differentiation is shifted to the late larva and even early pupa (*Makarova et al., 2022c*). According to the available data on the embryonic development of

Trichogrammatidae, no photoreceptor cells were found during the larval stages (*Ivanova-Kazas, 1954*; *Ivanova-Kazas, 1961*).

## Conclusion

Despite the extremely small body size, the compound eyes of *M. viggianii* retain an almost complete set of the cellular components of the ommatidia. The compound eye exhibits a regional specialization of ommatidia (DRA) putatively capable of polarized light perception. Ommatidia within the eye differ considerably in size and shape, and demonstrate corneal and retinal specializations. The results of the 3D reconstruction, morphometry, and volumetry of the key components of the ommatidia show a good match with the lamina connectivity patterns (*Chua et al., 2023*). A transitional zone is present between the adjacent non-DRA ommatidia of central area of the eye and DRA. Despite the nearly anucleate nervous system, the main sensory organs (such as the compound eye or Johnston's organ) of *M. viggianii* retain all their nuclei (*Diakova et al., 2022*). Our results not only reveal the general principles of the miniaturization of compound eyes but also provide context for future interpretation of the visual connectome of *M. viggianii*.

## Materials and methods

Adult females of *Megaphragma viggianii Polaszek et al., 2022* (Hymenoptera: Trichogrammatidae) were reared from eggs of *Heliothrips haemorrhoidalis* (Bouché, 1833) (Thysanoptera: Thripidae).

### FIB-SEM

Sample preparation was carried out according to a method described earlier (*Polilov et al., 2021*). The head was separated from the body in a cold fixative and immediately transferred to fresh fixative of 4°C for 1 h, which consisted of 1% glutaraldehyde (GA) and 1% osmium tetroxide ($OsO_4$) in 0.1 M sodium cacodylate buffer (pH = 7.2). The material was then washed in the same buffer and fixed for 2 h in 2% GA in the buffer at 4 °C. Next, the material was washed in the buffer and post-fixed for 16 h in 2% $OsO_4$ in the buffer at 4°C. After fixation material was washed with double distillate water, and then subjected to a 1% UA solution in $ddH_2O$ overnight at 4°C, and then placed (in the same solution) into a constant-temperature oven for 2 h at 50°C. The specimens were then washed in $ddH_2O$ and contrasted with Walton's lead aspartate solution (2 h, 50°C). The material was then washed in $ddH_2O$. Subsequently, dehydration of the material was continued using ethanol and acetone. The material was then placed in a mixture of an embedding medium (Epon, Sigma) and acetone (1 :2) for 2 h at room temperature (RT), and then in 1:1 mixture overnight at RT, after which the samples were transferred to a pouring medium for 5 h at RT. The samples were ultimately transferred to silicone embedding molds with fresh Epon and placed in a constant temperature oven for 48 h at 60°C.

The Epon embedded sample was mounted onto the top of a 1 mm copper stud using Durcupan, ensuring optimal charge dissipation by maintaining contact between the metal-stained sample and the copper stud. A thin layer of Durcupan resin was coated on the specimen front surface facing the FIB milling beam to mitigate the streaking artifacts caused by Epon resin.

The FIB-SEM prepared sample was imaged using a customized Zeiss NVision40 FIB-SEM system (*Xu et al., 2017*). The images were acquired using a 3 nA current SEM probe at 1.2 keV landing energy. Multiple imaging conditions were used to acquire the entire *M. viggianii* head. The pixel size along x and y axes was fixed at 8 nm. Scan rates were set to either 1.25 or 2.5 MHz, while z-steps of 2 nm or 4 nm were achieved by milling for 12–30 s with a 27 nA $Ga^+$ beam at 30 kV. A total volume of $64 \times 96 \times 98.6\ \mu m^3$ was acquired over the course of 90 days, spanning seven sections. The images were de-streaked using a MATLAB script, which applied a masked Fourier filter that removes the spatial frequencies corresponding to the streaks. The raw image stack was then aligned using a MATLAB script based on Scale Invariant Feature Transform (SIFT) and binned by a factor of 4 or 2 along the z-axis. The images were finally concatenated to create a dataset with $8 \times 8 \times 8\ nm^3$ voxels.

### SEM

The Bouin fixed material was gradually dehydrated through a series of ethyl alcohols 70%, 95% ethyl alcohol, each change for 30 min, 100% two changes for 30 min; and then acetone (100%, two changes

for 15 min), critical point dried (Hitachi HCP-2) and sputtered with gold (Giko IB-3). The specimens were studied and imaging was performed using Jeol JSM-6380 with a 5-megapixel digital camera.

## 3D reconstruction

The right eye was used for the 3D reconstruction, volumetric analysis, and morphometry. The right eye, on which the reconstruction was performed, has several damaged regions from milling (see *Figure 3—figure supplements 1–5*), which hinder the complete reconstruction of lenses and cones on a few ommatidia. According to this, for the volumetric data on lenses and cones, some linear measurements (lens thickness, cone length, cone width, curvature radius), we use (measure or reconstruct) the corresponding elements from the other (left) eye. The cells of single interfacet bristles were not reconstructed because of the damage present in the right eye and the generally lower quality of this region on the left eye.

All cellular and subcellular elements of the eye were manually segmented with Bitplane Imaris 9.5 on the right compound eye of the first specimen. The raw models were post-processed in Blender using smoothing and retopology tools. Volumes of cells and cell structures were calculated based on 3D models using the Imaris statistics module. Volumes of photoreceptor bodies were calculated without cell processes, as volumes of cone cells without their projections to the basement membrane. The pigment apparatus of the compound eye (the pigment granules) (*Figure 1F*) was reconstructed using the Ilastic software.

## Morphometry

Each ommatidium was numerated for comparison between ommatidia the same eye, and right and left eyes. All linear dimensions were measured on the FIB-SEM images using the measurement tools of Bitplane Imaris (v9.5). For rhabdomere segmentation the extra 29 stacks of ommatidia were performed from the FIB-SEM data using a Python script (N.J.C.) (see *Chua et al., 2023* 'Methods/ Optics measurements and calculations'). The rhabdomeres on each stack were segmented manually in Bitplane Imaris software. Statistical analysis was performed using STATISTICA 12, including *t*-tests for data analysis.

## Optical calculations

To compare the optical properties of the compound eyes, anatomical measurements were used to calculate relevant parameters: focal lengths, F-number, acceptance angle, and sensitivity of ommatidia (for a description of the formulas and parameters, see *Makarova et al., 2015*; see *Supplementary file 1d*).

## Identifying the retinula cells and terminology

For retinula cell numbering, we use the standardized *Drosophila*-based numbering convention (*Friedrich et al., 2011*) which is useful for understanding the photoreceptor subtype homologies. We combine cell body morphology (position of R8, basal cell) and axonal projection targets. The position of the eighth retinula cell in relation to the position of the cone cell projections (*Chua et al., 2023*) provides a means for the unique recognition and labeling of all other cells. The accuracy of identification of the three inner and six outer PRs was proved by projections of PR axons into lamina and medulla. Position and neuronal morphology data in *Megaphragma* lead to the conclusion that the extra inner PR also represents an R7' cell, nestled between the outer PR duet R3 and R4 and facing R7 along the medial axis of the ommatidium, as noted in *Friedrich et al., 2011*.

# Acknowledgements

This work was supported by the Russian Science Foundation (project no. 22-14-00028, AAP).

# Additional information

## Funding

| Funder | Grant reference number | Author |
|---|---|---|
| Russian Science Foundation | 22-14-00028 | Anastasia A Makarova<br>Anna V Diakova<br>Inna A Desyatirkina<br>Alexey A Polilov |

The funders had no role in study design, data collection and interpretation, or the decision to submit the work for publication.

## Author contributions

Anastasia A Makarova, Conceptualization, Data curation, Formal analysis, Investigation, Visualization, Methodology, Writing – original draft, Project administration, Writing – review and editing; Nicholas J Chua, Formal analysis, Investigation, Methodology, Writing – review and editing; Anna V Diakova, Inna A Desyatirkina, Writing – review and editing; Pat Gunn, Software, Methodology, Writing – review and editing; Song Pang, C Shan Xu, Methodology, Writing – review and editing; Harald F Hess, Resources, Supervision, Methodology, Project administration, Writing – review and editing; Dmitri B Chklovskii, Conceptualization, Supervision, Project administration, Writing – review and editing; Alexey A Polilov, Conceptualization, Resources, Supervision, Funding acquisition, Methodology, Project administration, Writing – review and editing

## Author ORCIDs

Anastasia A Makarova ⬚ https://orcid.org/0000-0001-8855-4128
Nicholas J Chua ⬚ https://orcid.org/0000-0002-6113-1296
Pat Gunn ⬚ https://orcid.org/0000-0003-1863-1081
Song Pang ⬚ https://orcid.org/0000-0002-7231-4151
C Shan Xu ⬚ https://orcid.org/0000-0002-8564-7836
Harald F Hess ⬚ https://orcid.org/0000-0003-3000-1533
Dmitri B Chklovskii ⬚ https://orcid.org/0000-0002-4781-2546
Alexey A Polilov ⬚ https://orcid.org/0000-0002-6214-3627

Reviewer #2 (Public review): https://doi.org/10.7554/eLife.103247.3.sa1
Reviewer #3 (Public review): https://doi.org/10.7554/eLife.103247.3.sa2
Author response https://doi.org/10.7554/eLife.103247.3.sa3

---

# Additional files

## Supplementary files

Supplementary file 1. Raw data for all measurements of ommatidia.

MDAR checklist

## Data availability

The vEM dataset analyzed during the current study is available on https://waspem-lamina.flatironinstitute.org/. The raw data of measurements available in *Supplementary file 1*.

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
