## [Editor Report · eLife Assessment]

This **valuable** study sets new standards in analyzing the ultrastructure of insect eyes, which have long served as models for understanding how vision works. The way it describes an entire eye with the resolution of electron microscopy is **convincing**. On top of this, a miniaturized visual system provides additional, remarkable insights towards understanding optimized solutions.

---

## [Referee Report · Reviewer #2 (Public review)]

Summary:

Makarova et al. provide the first complete cellular-level reconstruction of an insect eye. They use the extremely miniaturized parasitoid wasp, Megaphragma viggiani and apply improved and optimized volumetric EM methods they can describe, the size, volume and position of every single cell in the insect compound eye.

This data has previously only been inferred from TEM cross-sections taken in different parts of the eye, but in this study and in the associated 3d datasets video and stacks, one can observe the exact position and orientation in 3D space.

The authors have made a very rigorous effort to describe and assess the variation in each cell type and have also compared two different classes of dorsal rim and non-dorsal rim ommatidia and the associated visual apparatus for each, confirming previous known findings about the distribution and internal structure that assists in polarization detection in these insects.

Strengths:

The paper is well written and strives to compare the data with previous literature wherever possible and goes beyond cell morphology, calculating the optical properties of the different ommatidia and estimating light sensitivity and spatial resolution limits using rhabdom diameter, focal length and showing how this varies across the eye.

Finally, the authors provide very informative and illustrative videos showing how the cones, lenses, photoreceptors, pigment cells, and even the mitochondria are arranged in 3D space, comparing the structure of the dorsal rim and non-dorsal rim ommatidia. They also describe three 'ectopic' photoreceptors in more anatomical detail providing images and videos of them.

Comments on revisions:

The updates improve the manuscript.

---

## [Referee Report · Reviewer #3 (Public review)]

Summary:

The article presents a meticulous and quantitative anatomical reconstruction of the compound eye of a tiny wasp at the level of subcellular structures, cellular and optical organization of the ommatidia and reveals the ectopic photoreceptors, which are decoupled from the eye's dioptrical apparatus.

Strengths:

The graphic material is of very high quality, beautifully organized and presented in a logical order. The anatomical analysis is fully supported by quantitative numerical data at all scales, from organelles to cells and ommatidia, which should be a valuable source for future studies in cellular biology and visual physiology. The 3D renders are highly informative and a real eye candy.

Weaknesses:

The claim that the large dorsal part of the eye is the dorsal rim area (DRA), supported by anatomical data on rhabdomere geometry and connectomics in authors' earlier work, would eventually greatly benefit from additional evidence, obtained by other methods.

Comments on revisions:

Thank you for considering my remarks and advice. All is fine.

---

## [Author Response]

The following is the authors’ response to the original reviews

**Public reviews:**

**Reviewer #1:**
Weaknesses:As this paper only uses anatomical analyses, no functional interpretations of cell function are tested.The aim of this paper was to describe the ultrastructural organization of compound eyes in the extremely small wasp Megaphragma viggianii. The authors successfully achieved this aim and provided an incredibly detailed description of all cell types with respect to their location, volume, and dimensions. As this is the first of its kind, the results cannot easily be compared with previous work. The findings are likely to be an important reference for future work that uses similar techniques to reconstruct the eyes of other insect species. The FIB-SEM method used is being used increasingly often in structural studies of insect sensory organs and brains and this work demonstrates the utility of this method.

We thank you for your high assessment of our work. Unfortunately, it is hard to test our functional interpretations and check them with electrophysiological methods due to the extremely small size of the animal. Studies on three-dimensional ultrastructural datasets obtained using vEM have just started to appear, and we hope that a lot of data will become available for comparison in the nearest future.

**Reviewer #2:**

Thank you for your work and for your high assessment of our manuscript.

**Reviewer #3:**
Weaknesses:The claim that the large dorsal part of the eye is the dorsal rim area (DRA), supported by anatomical data on rhabdomere geometry and connectomics in authors' earlier work, would eventually greatly benefit from additional evidence, obtained by immunocytochemical staining, that could also reveal a putative substrate for colour vision. The cell nuclei that are located in the optical path in the DRA crystalline cone have only a putative optical function, which may be either similar to pore canals in hymenopteran DRA cornea (scattering) or to photoreceptor nuclei in camera-type eyes (focussing), both explanations being mutually exclusive.

We thank the Reviewer for high assessment of our study and for detailed analysis of our manuscript. Your comments and recommendations are very valued and helped us to improve the text. We understand that immunocytochemical methods could improve our findings and supply additional evidence, but there is no technical possibility for this in present. Megaphragma is a very complicated model organism for such methods. We are currently working on the optimization of the protocol for staining, which is needed because of the high level of autoluminescence and because of insufficient penetration of dyes into the samples.

**Recommendations for the authors:**

**Reviewer #1:**
I do not have any major concerns about the content of the paper.There are some minor spelling and grammatical errors throughout the text but these can be identified most readily using a spelling/grammar check.

We have revised the text, checked the spelling, and fixed the grammatical errors throughout the text.

I suggest consistency when referring to the capitalization of the term 'non-DRA' as it is sometimes 'Non-DRA' in the text.

We have fixed the term “non-DRA” throughout the text. Thank you.

Also, check carefully the spelling of headings in the tables as there are a few mistakes in Table 1 and 5 in particular.

The grammar errors have been fixed.

Figure 7 legend: an explanation of the abbreviation RPC should be added.

We have done so.

**Reviewer #2:**
(1) The paper presents the data in great detail, however, since this is the first time the technique has been applied to get whole insect eyes, even if on a small insect, it would be worth outlining in the methods section what innovations in the staining/ scanning or sample preparation allowed these improvements and a roadmap for extending this method to larger insects if possible.

The whole method, including sample preparation, staining, and scanning, was described in our previous paper (Polilov et al., 2021), where it was presented in every detail. Due to the complicated methodology we suppose that it is not necessary to include all the stages of the technique in the present paper, and thus described it more briefly.

(2) The optical modelling needs a statement in the discussion providing a disclaimer on parameters like sensitivity, anatomical measurements can provide limits and some measure, but the inherent optics are also key and it is worth qualifying these as only estimates and measurements that give a sense of the variation in morphology, only coupled with optical and potentially neural measurements could one confirm the true sensitivity and acceptance angle.

In the absence of experimental data or precise computational models of Megaphragma vision, we try to discuss rather carefully the functions of structures based on their morphology, ultrastructure, first-order visual connectome, and analogies with other species. This is reflected in the methods and those sections of our paper that contain functional interpretations.

**Reviewer #3**
(1) The finding that the CNS neurons are enucleated, while the compound eye contains cell nuclei, deserves another word. I would confidentially say that the optical demands of a miniaturized compound eye (the minimal size of the optics due to diffraction, the rhabdomere size, and the minimal thickness of optically insulating granules) are such that further cellular miniaturization is not possible, and the minimal sizes even render the cells that build the eye sufficiently large to accommodate cell nuclei. This is in my opinion a parsimonious explanation, yet speculative and I leave it up to you to embrace it or not.

We agree with the Reviewer and understand the limiting factors and the optical demands of a miniaturized compound eye. According to our data, nuclei occupy a considerable volume in the eye (in the cells of compound eye there are more nuclei than in the whole brain), and on average the cell volume is larger than in Trichogramma, which is minute, but larger than Megaphragma. But as the Reviewer rightly assumed, it is speculative; therefore, we would like to avoid it.

(2) Our current understanding of DRA optics and function is limited and I claim that your interpretation of the cell nuclei in the DRA dioptrical apparatuses is inappropriate. Please consider a few articles on hymenopteran DRA, starting with the one below and the citing literature:Meyer, E.P., Labhart, T. Pore canals in the cornea of a functionally specialized area of the honey bee's compound eye. Cell Tissue Res. 216, 491-501 (1981). https://doi.org/10.1007/BF00238646Honebyee DRA has a milky appearance under a stereomicroscope and can be discerned from the outside. This is due to pore canals in the cornea. I happen to be studying this exact structure and its function right now. I found that the result of those canals is not so much the extended receptor acceptance angles, but rather a minimized light gain. This is counterintuitive, but think of the following. The DRA photoreceptors must encode the limited range of polarization contrasts with a maximal working dynamic range (=voltage) of the photoreceptors, which results in a very steep stimulus-response curve.Physiologically such a curve is due to very high transduction gain and a high cell input resistance. In most of the retina, small contrasts are transcoded by LMC neurons, but DRA receptors are long visual fibres and must do the job themselves. The skylight intensity (especially antisolar, where the polarized pattern is maximal) varies little during the day. Hence, the DRA receptors work almost at a fixed intensity range. In order to prevent receptor saturation and keep steep contrast coding, the corneal lenses in DRA have a built-in diffusor ring, which diminishes the light influx. Unfortunately, I have yet to publish this and I may be wrong, of course. But if I look into your data, I see consistently smaller corneal lenses and crystalline cones in the DRA, plus the cell nuclei obstructing the incident light. I think this is similar to the optics of honeybee DRA.You do not support your claim that the nuclei additionally focus light by optical calculations, but cite literature on camera-type eyes, which is not OK.In any case, I think it is fair to limit the discussion by saying that the nuclei may have an optical role. Further evidence from hymenopteran and vertebrate literature is controversial. “so that the nuclei act as extra collecting lenses, as was reported for rod cells of nocturnal vertebrates (Solovei et al., 2009; Błaszczak et al., 2014)” - please consider omitting this.

We thank the Reviewer for this piece of advice. And we have rewritten the text, to omit the comparison with vertebrates, but left the citation as an illustration of the fact that nuclei could perform the optical role.

“Since the nuclei in DRA and non-DRA ommatidia are arranged differently in cone cells, we suggest that the nuclei of the cone cells of DRA ommatidia in M. viggianii perform some optical role, facilitating the specialization of this group of ommatidia. The optical function for nuclei was described for rod cells of nocturnal vertebrates, where chromatin inside the cell nucleus has a direct effect on light propagation (Solovei et al., 2009; Błaszczak et al., 2014; Feodorova et al., 2020).”

(3) Please consider comparing the structure and function of ectopic receptors with the eyelet in *Drosophila* (i.e. https://doi.org/10.1523/JNEUROSCI.22-21-09255.2002)

We thank the Reviewer for this advice and have included the comparison fragment into the text:

“The position of ePR, their morphology and synaptic targets look similar to the eyelet (extraretinal photoreceptor cluster) discovered in *Drosophila* (Helfrich-Förster et al., 2002). Eyelets are remnants of the larval photoreceptors, Bolwig’s organs in *Drosophila* (Hofbauer, Buchner, 1989). Unlike *Drosophila*, Trichogrammatidae are egg parasitoids and their central nervous system differentiation is shifted to the late larva and even early pupa (Makarova et al., 2022). According to the available data on the embryonic development of Trichogrammatidae, no photoreceptors cells were found during the larval stages (Ivanova-Kazas, 1954, 1961).”

According to this, the analogy question remains open.

(4) Minor remarks:“but also to trace the pathways that connect the analyzer with the brain.” - I find the word analyzer a bit stretched here; sure, the DRA is polarization analyzer, but if the main retina was monochromatic, it would only be a detector, not an analyzer.

The sentence was changed according to the Reviewer’s advice.

Table I: thikness -> thickness, wigth -> width

We have fixed these misprints.

“The cross-section of Non-DRA ommatidia has a strongly spherical shape” - perhaps circular, not spherical. And not necessary to say “strongly”

The spelling was changed according to the Reviewer’s advice.

“which can be rarely visualized in the cell's projections not far from the basement membrane.” - I'd suggest saying “which are nearly absent in retinula axons”

The spelling was changed according to the Reviewer’s advice.

“The pigment granules of the retinula cells have an elongated nearly oval shape” - please consider replacing 'elongated nearly oval' with 'prolate' (try googling for “prolate” or “oblate spheroids”; the adjective describes precisely what you wanted to say)

We thank the Reviewer for this piece of advice but prefer to leave our original phrasing, because it is more readily understandable.

“The results of our morphological analysis of all ommatidia in Megaphragma are consistent with the light-polarization related features in Hymenoptera and other insects” - please add citations, see my comment on the DRA above.

We have added the citations according to the Reviewer’s advice.

“The group of short PRs (R1-R6)” - please consider renaming into “short visual fibre photoreceptors” (as opposed to “long visual fibre PRs”; hence SVFs and LVFs). This naming is quite common.

The naming was changed according to the Reviewer’s advice.

“The total rhabdom shortening in M. viggianii ommatidia probably favors polarization and absolute sensitivity,” - please see comments on DRA. Wide rhabdom means also a wider acceptance angle.

Shortening of DRA rhabdoms does not result in their widening compared to other rhabdoms, so it is difficult to say how this may be related to sensitivity. The comments on DRA given earlier have been taken into account.

“Ommatidia located across the diagonal area of the eye are more sensitive to light” - I don't understand what is diagonal area.

We have deleted the sentence.

“Estimated optical sensitivity of the eyes very close to those reported for diurnal hymenopterans with apposition eyes (Greiner et al., 2004; Gutiérrez et al., 2024) and possess around 0.19 {plus minus} 0.04 μm2 sr. M. viggianii have reasonably huge values of acceptance angle Δρ, and thus should result in a low spatial resolution” - please correct English here. “eyes IS very close”, “should result in a low”

The grammatical errors were fixed.

Table 6 legend: “SPC - secondary pigment cells.” -> “SPC – secondary pigment cells.”Citation “(Makarova et al., 2025).” - probably 2015

The typos were fixed.

Methods, FIB-SEM: I can't understand the sentence “The volumetric data of lenses and cones, some linear measurements (lens thickness, cone length, cone width, curvature radius) and to visualize the complete 3D-model of eye we use (measure or reconstruct) the elements from another eye (left).”

The sentence is a continuation of the previous one. We have rewritten it as follows to clarify the meaning and move it to the 3D reconstruction section:

“The right eye, on which the reconstruction was performed, has several damaged regions from milling (see Appendix 1С), which hinder the complete reconstructions of lenses and cones on a few ommatidia. According to this, for the volumetric data on lenses and cones, some linear measurements (lens thickness, cone length, cone width, curvature radius), we use (measure or reconstruct) the corresponding elements from the other (left) eye.”

“The cells of single interfacet bristles were not reconstructed, because of damaging on right eye and worst quality of section on the left.” - please change to “The cells of the single interfacet bristle were not reconstructed, because of damage to the right eye and inferior quality of the sections of the left eye.”

The text has been changed as follows:

“The cells of single interfacet bristles were not reconstructed, because of the damage present in the right eye and because of the generally lower quality of this region on the left eye.”

“Morphometry. Each ommatidia was” -> “Morphometry. Each ommatidium was”

The grammatical error has been fixed.